# A generative model for inorganic materials design

Claudio Zeni[1,8], Robert Pinsler[1,8], Daniel Zügner[2,8], Andrew Fowler[1,8], Matthew Horton[3,8], Xiang Fu[1], Zilong Wang[4], Aliaksandra Shysheya[1], Jonathan Crabbé[1], Shoko Ueda[1], Roberto Sordillo[1], Lixin Sun[1], Jake Smith[3], Bichlien Nguyen[3], Hannes Schulz[2], Sarah Lewis[1], Chin-Wei Huang[5], Ziheng Lu[6], Yichi Zhou[7], Han Yang[6], Hongxia Hao[6], Jielan Li[6], Chunlei Yang[4], Wenjie Li[4], Ryota Tomioka[1,8 ✉] & Tian Xie[1,8 ✉]

The design of functional materials with desired properties is essential in driving technological advances in areas such as energy storage, catalysis and carbon capture[1–3]. Generative models accelerate materials design by directly generating new materials given desired property constraints, but current methods have a low success rate in proposing stable crystals or can satisfy only a limited set of property constraints[4–11]. Here we present MatterGen, a model that generates stable, diverse inorganic materials across the periodic table and can further be fine-tuned to steer the generation towards a broad range of property constraints. Compared with previous generative models[4,12], structures produced by MatterGen are more than twice as likely to be new and stable, and more than ten times closer to the local energy minimum. After fine-tuning, MatterGen successfully generates stable, new materials with desired chemistry, symmetry and mechanical, electronic and magnetic properties. As a proof of concept, we synthesize one of the generated structures and measure its property value to be within 20% of our target. We believe that the quality of generated materials and the breadth of abilities of MatterGen represent an important advancement towards creating a foundational generative model for materials design.

The rate at which we can discover better materials has a substantial impact on the pace of technological innovation in areas such as carbon capture, semiconductor design and energy storage[1–3]. Traditionally, most materials have been discovered through experimentation and human intuition, limiting the number of candidates that can be tested and causing long iteration cycles. Owing to the advance of high-throughput screening[13], open material databases[14–17], machine-learning-based property predictors[18,19] and machine learning force fields (MLFFs)[20,21], it has become possible to screen hundreds of thousands of materials to identify promising candidates[22,23]. However, screening-based methods are still fundamentally limited by the number of known materials. The largest explorations of previously unknown crystalline materials are in the orders of $10^6$–$10^7$ materials[21,23–25], which is only a tiny fraction of the number of potentially stable inorganic compounds[26]. Moreover, these methods cannot be efficiently steered towards finding materials with target properties.

Given these limitations, there has been great interest in the inverse design of materials[27,28]. The aim of inverse design is to directly generate material structures that satisfy target property constraints, for example, using generative models[4,8,11], evolutionary algorithms[29] and reinforcement learning[30]. Generative models are promising because they can efficiently explore new structures and be flexibly adapted to different downstream tasks. However, current generative models often fall short of producing stable materials according to density functional theory (DFT) calculations[4,5,31], are constrained by a narrow subset of elements[7,9] and/or can only optimize a very limited set of properties, mainly formation energy[4,5,8,11,31,32].

In this study, we present MatterGen, a diffusion-based generative model that generates stable, diverse inorganic materials across the periodic table and can be fine-tuned towards a wide range of downstream tasks for inverse materials design (Fig. 1). To enable this, we introduce a diffusion process that generates crystal structures by gradually refining atom types, coordinates and the periodic lattice. We further introduce adapter modules to enable fine-tuning on desired chemical composition, symmetry and scalar property constraints such as magnetic density. Compared with previous state-of-the-art generative models for materials[4,12], MatterGen more than doubles the percentage of generated stable, unique and new (SUN) materials and generates structures that are more than ten times closer to their ground-truth structures at the DFT local energy minimum (Fig. 2). The broad conditioning abilities of MatterGen enable inverse materials design for a much wider range of problems than previous generative models. When fine-tuned, MatterGen often generates more SUN materials in target chemical systems than well-established methods such as substitution and random structure search (RSS) (Fig. 3), can generate highly symmetric structures given desired space groups (Fig. D8)

[1]Microsoft Research AI for Science, Cambridge, UK. [2]Microsoft Research AI for Science, Berlin, Germany. [3]Microsoft Research AI for Science, Redmond, WA, USA. [4]Shenzhen Institute of Advanced Technology, Chinese Academy of Sciences, Shenzhen, China. [5]Microsoft Research AI for Science, Amsterdam, The Netherlands. [6]Microsoft Research AI for Science, Shanghai, China. [7]Microsoft Research AI for Science, Beijing, China. [8]These authors contributed equally: Claudio Zeni, Robert Pinsler, Daniel Zügner, Andrew Fowler, Matthew Horton, Ryota Tomioka, Tian Xie. ✉e-mail: ryoto@microsoft.com; tianxie@microsoft.com

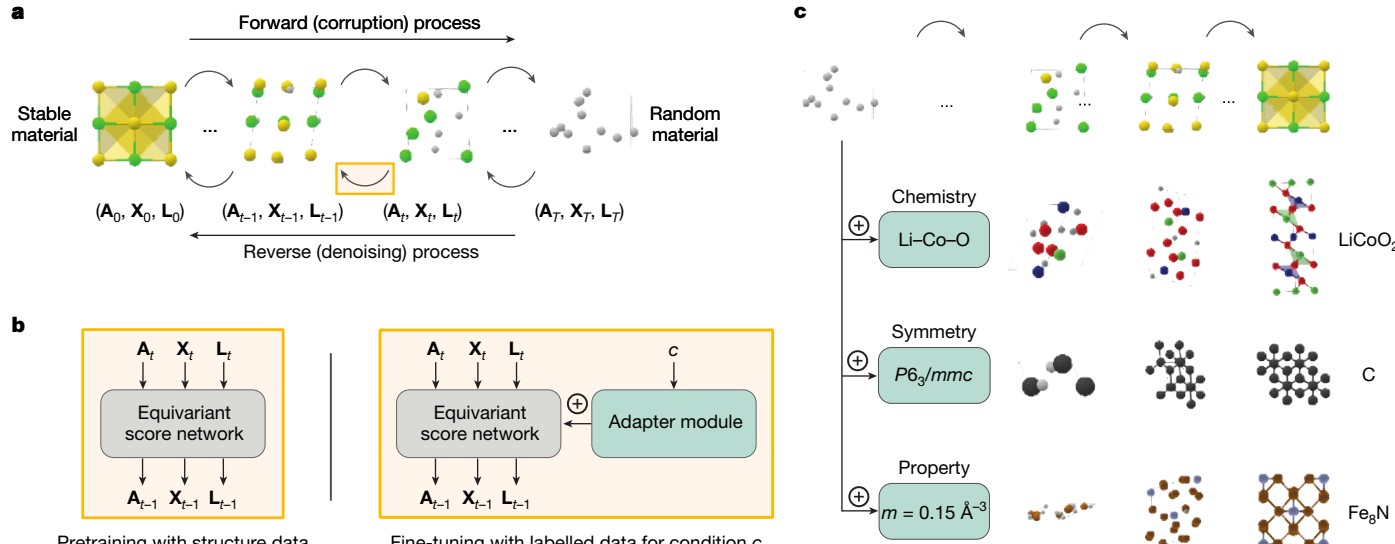

**Fig. 1 | Inorganic materials design with MatterGen. a**, MatterGen generates stable materials by reversing a corruption process through iteratively denoising a random structure. The forward diffusion process independently corrupts atom types **A**, coordinates **X** and the lattice **L** towards a physically motivated distribution of random materials. **b**, An equivariant score network is pretrained on a large dataset of stable material structures to jointly denoise atom types, coordinates and the lattice. The score network is then fine-tuned with a labelled dataset through an adapter module that adapts the model using the encoded property *c*. **c**, The fine-tuned model generates materials with desired chemistry, symmetry or scalar property constraints. *m*, magnetic density.

and directly generate SUN materials that satisfy target mechanical, electronic and magnetic property constraints (Fig. 4). MatterGen is also able to design materials given multiple property constraints, for example, high magnetic density and chemical composition with low supply-chain risk (Fig. 5). As a proof of concept, we validate the design abilities of MatterGen by synthesizing a generated material and measuring its property to be within 20% of our target (Fig. 6).

## Diffusion process for materials

MatterGen is a diffusion model tailored for designing crystalline materials across the periodic table (Fig. 1a). Diffusion models generate samples by reversing a fixed corruption process using a learned score network[33–35]. Corruption processes for images typically add Gaussian noise but crystalline materials have unique periodic structure and symmetries that demand a customized diffusion process. We define a crystalline material by its repeating unit, that is, its unit cell, comprising the atom types **A** (that is, chemical elements), coordinates **X** and periodic lattice **L** (Supplementary Information sections A.1 and A.2). For each component, we define a corruption process that considers its particular geometry and has a physically motivated limiting noise distribution. The coordinate diffusion respects the periodic boundary using a wrapped Normal distribution and approaches a uniform distribution at the noisy limit. We adjust for the effect of cell size on the fractional coordinate diffusion in Cartesian space by scaling the noise magnitude accordingly (Supplementary Information section A.6). Our lattice diffusion takes a symmetric form and approaches a distribution whose mean is a cubic lattice with average atomic density from the training data (Supplementary Information section A.7). Atom types are diffused in categorical space in which individual atoms are corrupted into a masked state (Supplementary Information section A.5). To reverse the corruption process, we learn a score network that outputs invariant scores for atom types and equivariant scores for coordinates and lattice, removing the need to learn symmetries from data (Supplementary Information sections A.8 and A.9).

To design materials with desired property constraints, we introduce adapter modules for fine-tuning the score model on an additional dataset with property labels (Fig. 1b and Supplementary Information section B). The adapter modules are tunable components injected into each layer of the base model to alter its output depending on the given property label[36]. Fine-tuning is appealing as it still works well if the labelled dataset is small compared with unlabelled structure datasets, as is often the case owing to the high computational cost of calculating properties. The fine-tuned model is used in combination with classifier-free guidance[37] to steer the generation towards target property constraints. We apply this approach to multiple types of constraints, producing a set of fine-tuned models that can generate materials with target chemical composition, symmetry or scalar properties such as magnetic density (Fig. 1c). These broad conditioning abilities combined with the improvements in the diffusion process over previous work[4,12] are key for addressing a wide range of inverse design problems (Supplementary Information section A.11).

## Generating stable, diverse materials

We formulate learning a generative model for inverse materials design as a two-step process, in which we first pretrain a general base model for generating stable, diverse crystals across the periodic table and then we fine-tune this model towards different downstream tasks. To train the base model, we curate a large and diverse dataset, Alex-MP-20, comprising 607,683 stable structures with up to 20 atoms recomputed from the Materials Project (MP)[14] and Alexandria[25,38] datasets (Supplementary Information section C).

In this section, we focus on the ability of the base model of MatterGen to generate stable, diverse materials, which we argue is a prerequisite for addressing any inverse materials design task. Since diversity is difficult to measure directly, we resort to quantifying the ability of MatterGen to generate SUN materials (Supplementary Information section D.3) and provide further analysis of the quality and diversity of generated structures. We consider a structure to be stable if its energy per atom after relaxation via DFT is within 0.1 eV per atom above the convex hull defined by a reference dataset, Alex-MP-ICSD, comprising 850,384 unique structures recomputed from the MP[14], Alexandria[25,38] and Inorganic Crystal Structure Database (ICSD)[39] datasets (Supplementary Information section C). We consider a structure

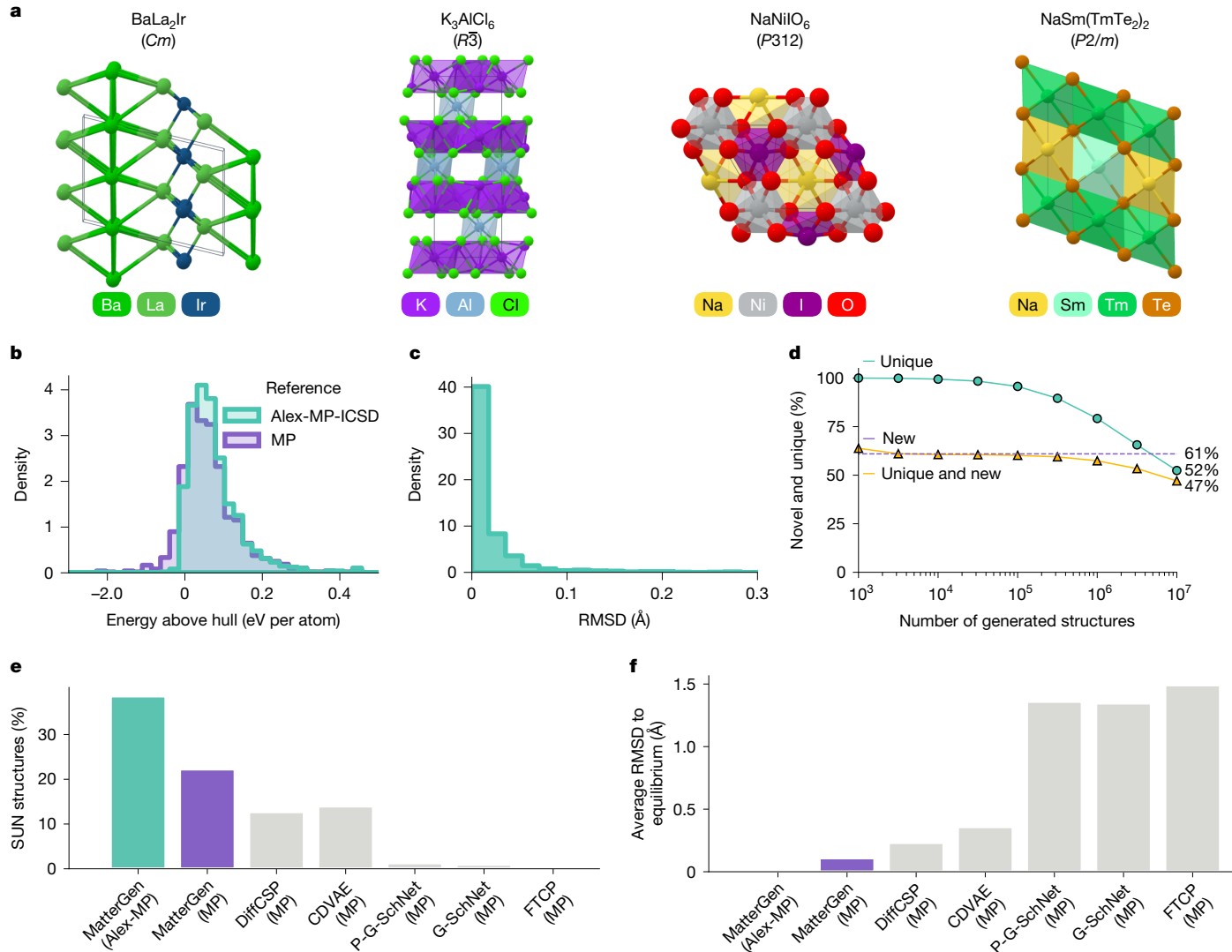

**Fig. 2 | Generating stable, unique and new inorganic materials.**
**a**, Visualization of four randomly selected crystals generated by MatterGen, with corresponding reduced formula and space group. **b**, Distribution of energy above hull values of generated structures using MP and Alex-MP-ICSD datasets as energy references, respectively. **c**, Distribution of root mean squared displacement (RMSD) between initial generated and DFT-relaxed structures. **d**, Percentage of unique, new structures as a function of the number of generated structures. **e**,**f**, Percentage of SUN structures (**e**) and average RMSD between initial and DFT-relaxed structures (**f**) for MatterGen, MatterGen-MP and several baseline models, including DiffCSP[12], CDVAE[4], P-G-SchNet, G-SchNet[50] and FTCP[31]. Training datasets are in parentheses. Percentage of SUN structures are computed using 1,024 samples for MatterGen and 1,000 for baseline models.

to be unique if it does not match any other structure generated by the same method. We consider a structure to be new if it does not match any structure present in an extended version of Alex-MP-ICSD containing 117,652 disordered ICSD structures in addition to the 850,384 ordered structures used to compute the reference convex hull. To account for compositional disorder effects[40], we match structures based on a newly proposed ordered-disordered structure matcher (Supplementary Information section D.4). We adopt these definitions throughout unless stated otherwise.

Figure 2a shows several random samples generated by MatterGen, featuring typical coordination environments of inorganic materials (see Supplementary Information section D.5.3 for a more detailed analysis). To assess stability, we perform DFT calculations on 1,024 generated structures. Figure 2b shows that 78% of generated structures fall below the 0.1 eV per atom threshold (13% below 0 eV per atom) of the convex hull of MP, whereas 75% fall below the 0.1 eV per atom threshold (3% below 0 eV per atom) of the combined Alex-MP-ICSD

hull. Furthermore, 95% of generated structures have an RMSD with respect to their DFT-relaxed structures that is below 0.076 Å (Fig. 2c), which is almost one order of magnitude smaller than the atomic radius of the hydrogen atom (0.53 Å). These results indicate that most of the structures generated by MatterGen are stable and very close to the DFT local energy minimum.

We further investigate whether MatterGen can generate a substantial amount of unique and new materials. We find that the percentage of unique structures is 100% when generating 1,000 structures and only drops to 52% after generating 10 million structures, whereas 61% of generated structures are new (Fig. 2d). This suggests that MatterGen can generate diverse structures without significant saturation even at a large scale and that most of those structures are new with respect to Alex-MP-ICSD. Remarkably, we also find that MatterGen has rediscovered more than 2,000 experimentally verified structures from ICSD not seen during training (Supplementary Information section D.5.4), showing its ability to generate synthesizable materials.

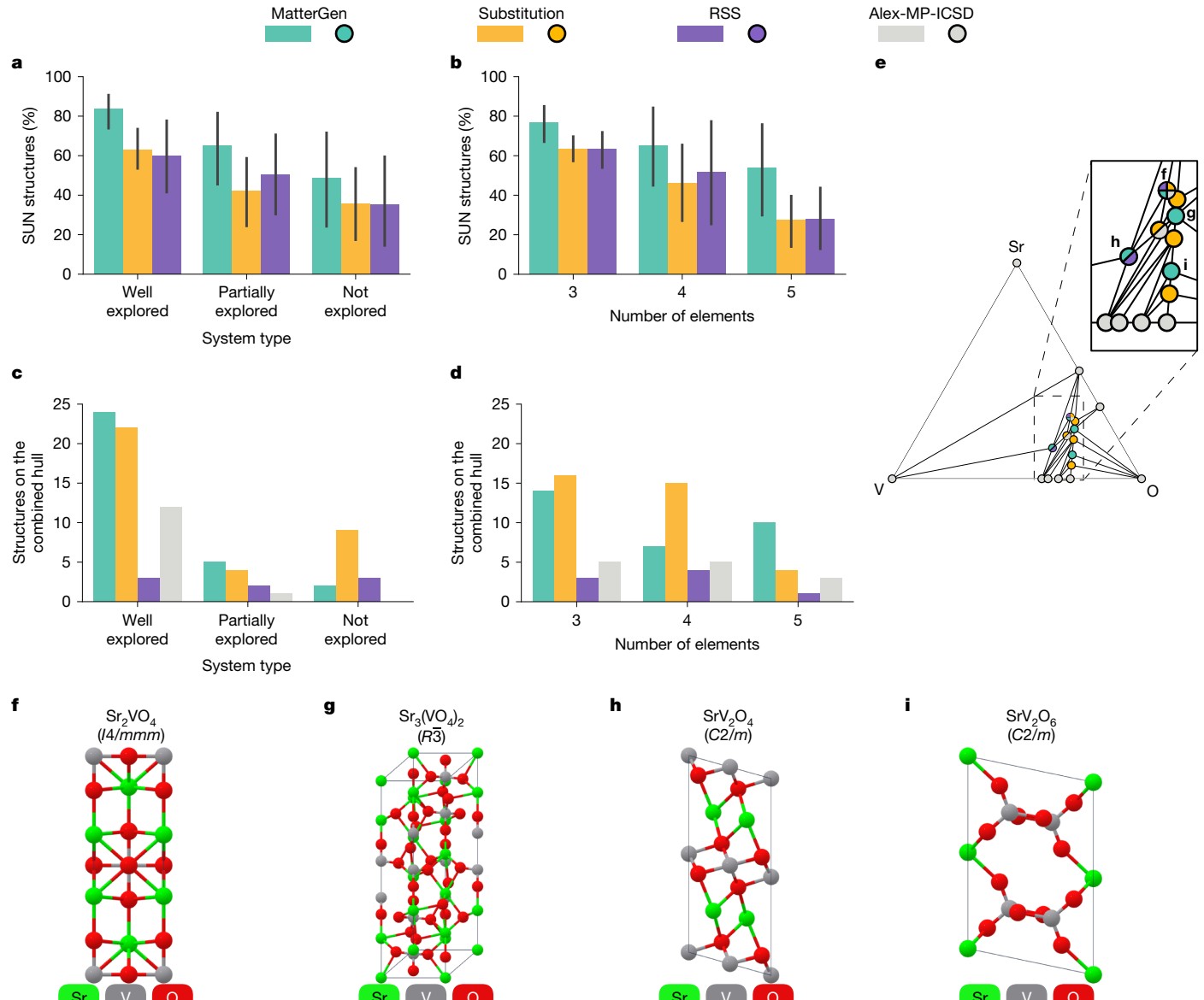

**Fig. 3 | Generating materials in target chemical system. a,b,** Mean percentage of SUN structures generated by MatterGen and baselines for 27 chemical systems, grouped by system type (**a**) and number of elements (**b**). Percentages are computed on 100 structures for each of 9 chemical systems. Error bars denote 95% percentile intervals ($n = 9$). **c,d,** Number of structures on the combined convex hull found by each method and in the Alex-MP-ICSD dataset, grouped by system type (**c**) and number of elements (**d**). **e,** Convex hull diagram for V–Sr–O, a well-explored ternary system. Dots denote structures on the hull, their coordinates show the element ratio of their composition and their colour indicates by which method they were discovered. **f–i,** Four structures that MatterGen discovered (rediscovered in the case of **f**) on the V–Sr–O hull shown in **e,** along with their reduced formula: $Sr_2VO_4$ (**f**), $Sr_3(VO_4)_2$ (**g**), $SrV_2O_4$ (**h**) and $SrV_2O_6$ (**i**).

Next, we benchmark MatterGen against previous generative models for materials and show a substantial performance improvement. We focus on two metrics averaged over 1,000 generated samples from each method: (1) the percentage of SUN materials among generated samples, measuring the success rate of generating promising candidates and (2) the average RMSD between generated samples and their DFT-relaxed structures, measuring the distance to equilibrium (Supplementary Information section D.5.1). We also compare with MatterGen-MP, which is a MatterGen model trained only on MP-20, that is, the same, smaller, dataset used by the other baselines. Compared with the previous state-of-the-art methods CDVAE[4] and DiffCSP[12], MatterGen-MP generates 60% more SUN structures whereas the average RMSD of the generated structures is 50% lower (Fig. 2e,f). We find that our model design choices are crucial for the improved performance (Supplementary Information section A.10). When comparing MatterGen with

MatterGen-MP, we observe a further 70% increase in the percentage of SUN structures and a five times decrease in RMSD as a result of scaling up the training dataset.

Combining both model and data improvements, MatterGen generates structures that are more than twice as likely to be SUN compared with previous generative models, whereas the generated structures are up to an order of magnitude closer to their local energy minimum. Next, we fine-tune the pretrained base model of MatterGen towards different downstream applications, including target chemistry (see section 'Chemistry-guided design') and scalar property constraints (see sections 'Property-guided design' and 'Designing low-supply-chain-risk magnets'), with experimental validation in the section 'Experimental validation'. Results for fine-tuning on symmetry constraints are in Supplementary Information section D.7.

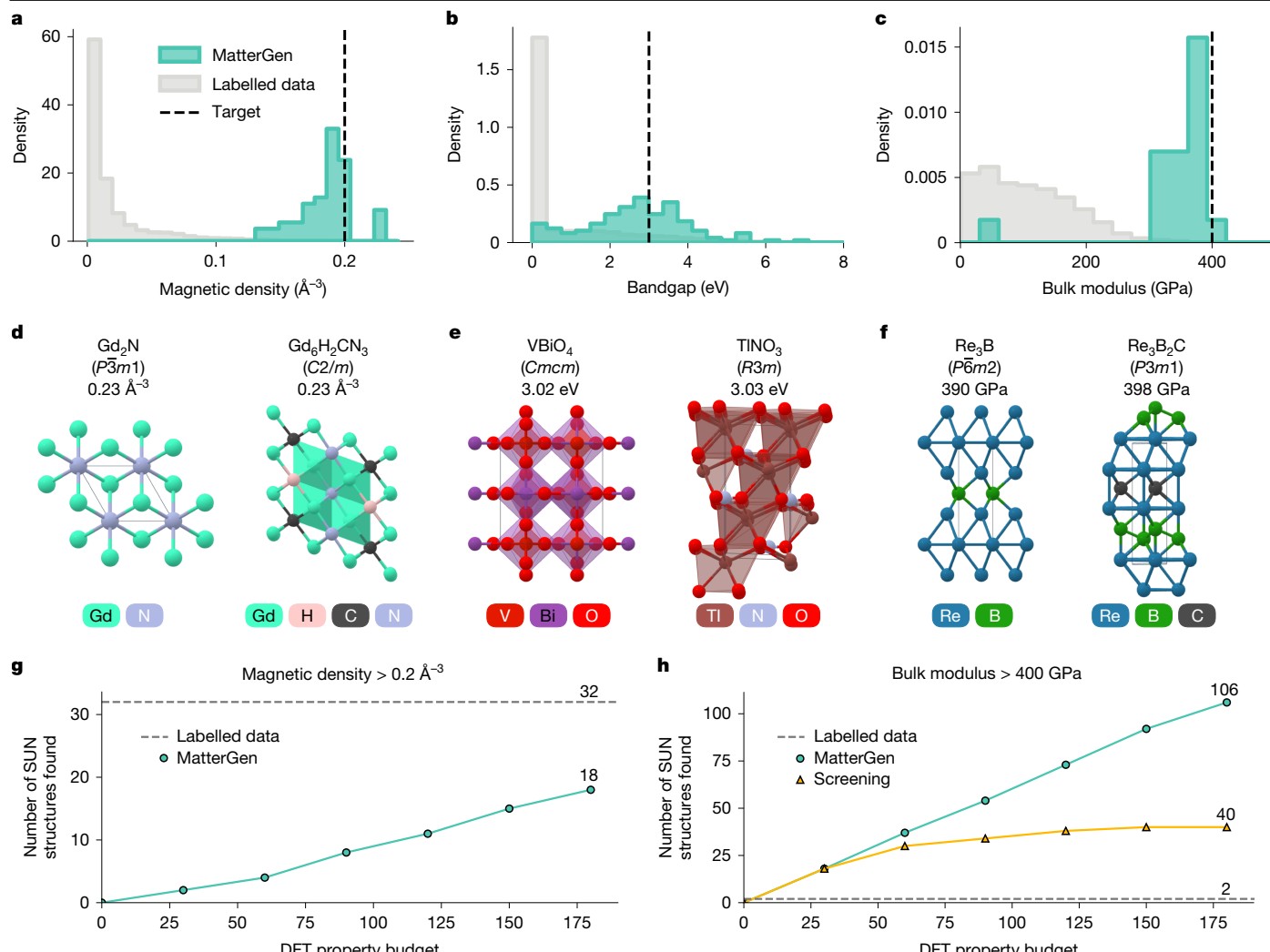

**Fig. 4 | Designing materials with target magnetic, electronic and mechanical properties. a–c**, Density of property values among (1) SUN samples generated by MatterGen and (2) structures in the labelled fine-tuning dataset for magnetic (**a**), electronic (**b**) and mechanical (**c**) properties. The property target for MatterGen is shown as a black dashed line. Magnetic density values less than $10^{-3}$ Å$^{-3}$ in **a** are excluded from the labelled data to improve readability. **d–f**, Visualization of SUN structures with the best property values generated by MatterGen for magnetic density (**d**), bandgap (**e**) and bulk modulus (**f**), along with their reduced formula, space group and property value. **g,h**, Number of SUN structures that satisfy target constraints found by MatterGen and baselines across DFT property calculation budgets: magnetic density > 0.2 Å$^{-3}$ (**g**) and bulk modulus > 400 GPa (**h**).

## Chemistry-guided design

Finding the most stable material structures in a target chemical system (for example, Li–Co–O) is crucial to define the true convex hull required for assessing stability and is one of the main challenges in materials design[41]. The most comprehensive approach for this task is ab initio RSS[42], which has been used to discover many new materials that were later experimentally synthesized[41]. The biggest drawback of RSS is its computational cost, as the thorough exploration of even a ternary compound can require hundreds of thousands of DFT relaxations. In recent years, the combination of generating structures by RSS, substitution or evolutionary methods with MLFFs has proven successful in exploring chemical systems[21,23,43].

Here we evaluate the ability of MatterGen to explore target chemical systems by comparing it with substitution and RSS. We equip all methods with the MatterSim[44] MLFF to pre-relax and filter the generated structures by their predicted stability before running more expensive DFT calculations. We fine-tune the MatterGen base model (Supplementary Information section B.1) and steer the generation towards different target chemical systems and an energy above hull of 0 eV per atom. We evaluate the methods on nine ternary, nine quaternary and nine quinary chemical systems. For each of these three groups, we pick three chemical systems at random from the following categories: well explored, partially explored and not explored (Supplementary Information section D.6).

MatterGen generates the highest percentage of SUN structures for every system type and every chemical complexity (Fig. 3a,b). Moreover, MatterGen finds the highest number of unique structures on the combined convex hull in (1) partially explored systems, in which the existing known structures near the hull were provided during training; and (2) well-explored systems, in which the structures near the hull are known but were not provided in training (Fig. 3c). Although substitution offers a comparable or more efficient way to generate structures on the hull for ternary and quaternary systems, MatterGen achieves better performance on quinary systems (Fig. 3d). Remarkably, the strong performance of MatterGen in quinary systems was achieved with only 10,240 generated samples, compared with about 70,000 samples for substitution and 600,000 for RSS. This underscores the enormous efficiency gains that can be realized with generative models by proposing better initial candidates. Finally, we show that MatterGen finds three

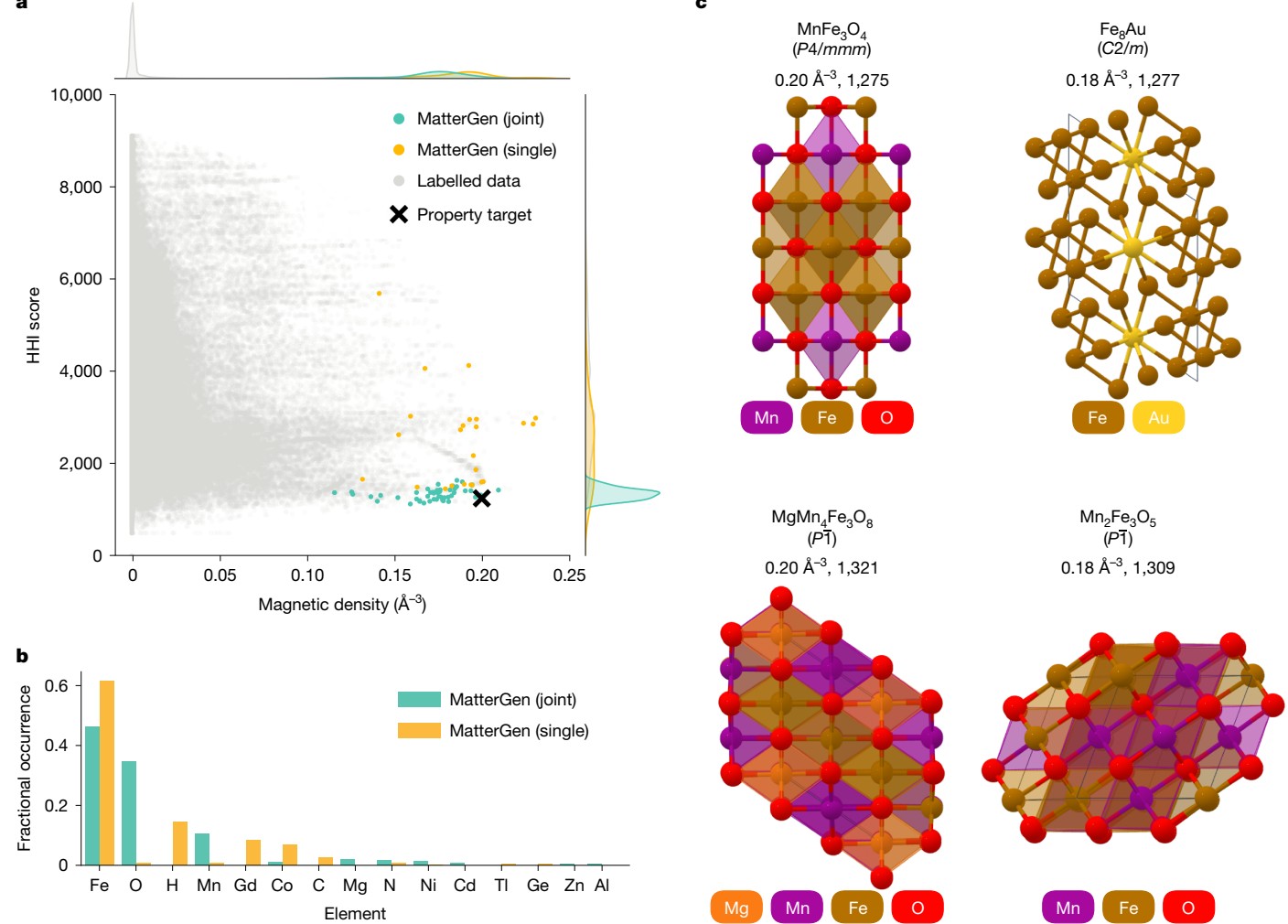

**Fig. 5 | Designing low-supply-chain-risk magnets. a**, Distribution of SUN structures generated by MatterGen when fine-tuned on magnetic density (single) and on both HHI score and magnetic density (joint), as well as structures from the labelled fine-tuning dataset. The property target of MatterGen is shown as a black cross. **b**, Occurrence of most frequent elements in SUN structures for the two fine-tuned MatterGen models. **c**, SUN structures on the Pareto front for the jointly fine-tuned model, along with their reduced formula, space group, magnetic density and HHI score.

## Property-guided design

There is an enormous need for materials with improved properties across many applications, including energy storage, catalysis and carbon capture[1–3]. The classical screening-based approach starts from a set of candidates and selects the ones with the best-predicted properties, but screening cannot explore structures beyond the set of known materials. Here we demonstrate the ability of MatterGen to directly generate SUN materials with target constraints on three different inverse design tasks, featuring a diverse set of properties—magnetic, electronic and mechanical—with varying degrees of available labelled data for fine-tuning the model. In the first task, we aim to generate materials with high magnetic density, a prerequisite for permanent magnets. We fine-tune the model on 605,000 structures with DFT magnetic density labels (calculated assuming ferromagnetic ordering) and generate structures with a target magnetic density value of 0.20 Å$^{-3}$. Second, we fine-tune the model on 42,000 structures with DFT bandgap labels

and sample materials with a target bandgap value of 3.0 eV. Finally, we target structures with high bulk modulus—an important property for superhard materials. We fine-tune the model on only 5,000 labelled structures and sample with a target value of 400 GPa. Although these tasks were chosen to evaluate the generality of the model, further investigations would be required to assess the suitability of these materials for specific applications. For example, a superhard material needs to have a high shear modulus, and a permanent magnet needs a suitable magnetic order and critical temperature. Further experimental details are in Supplementary Information section D.8.

In Fig. 4a–c, we observe a substantial shift in the distribution of property values among SUN samples generated by MatterGen towards the desired targets, even when the targets are at the tail of the data distribution. This still holds true for properties for which the number of DFT labels available for fine-tuning the model is substantially smaller than the size of the unlabelled training data. In Fig. 4d–f, we show the SUN structures with the best-predicted property values generated by MatterGen for each task, with further analysis in Supplementary Information section D.8.2.

Moreover, we assess the number of SUN structures satisfying extreme property constraints that can be found by MatterGen when given a limited budget for DFT property calculations. As a baseline, we count the number of materials in the labelled fine-tuning dataset that satisfy

new (four overall) structures on the combined hull for V–Sr–O—an example of a well-explored ternary system—whereas substitution finds three (five overall) and RSS only one (two overall) (Fig. 3e). Structures discovered by MatterGen are shown in Fig. 3f–i and are analysed in Supplementary Information section D.6.2.

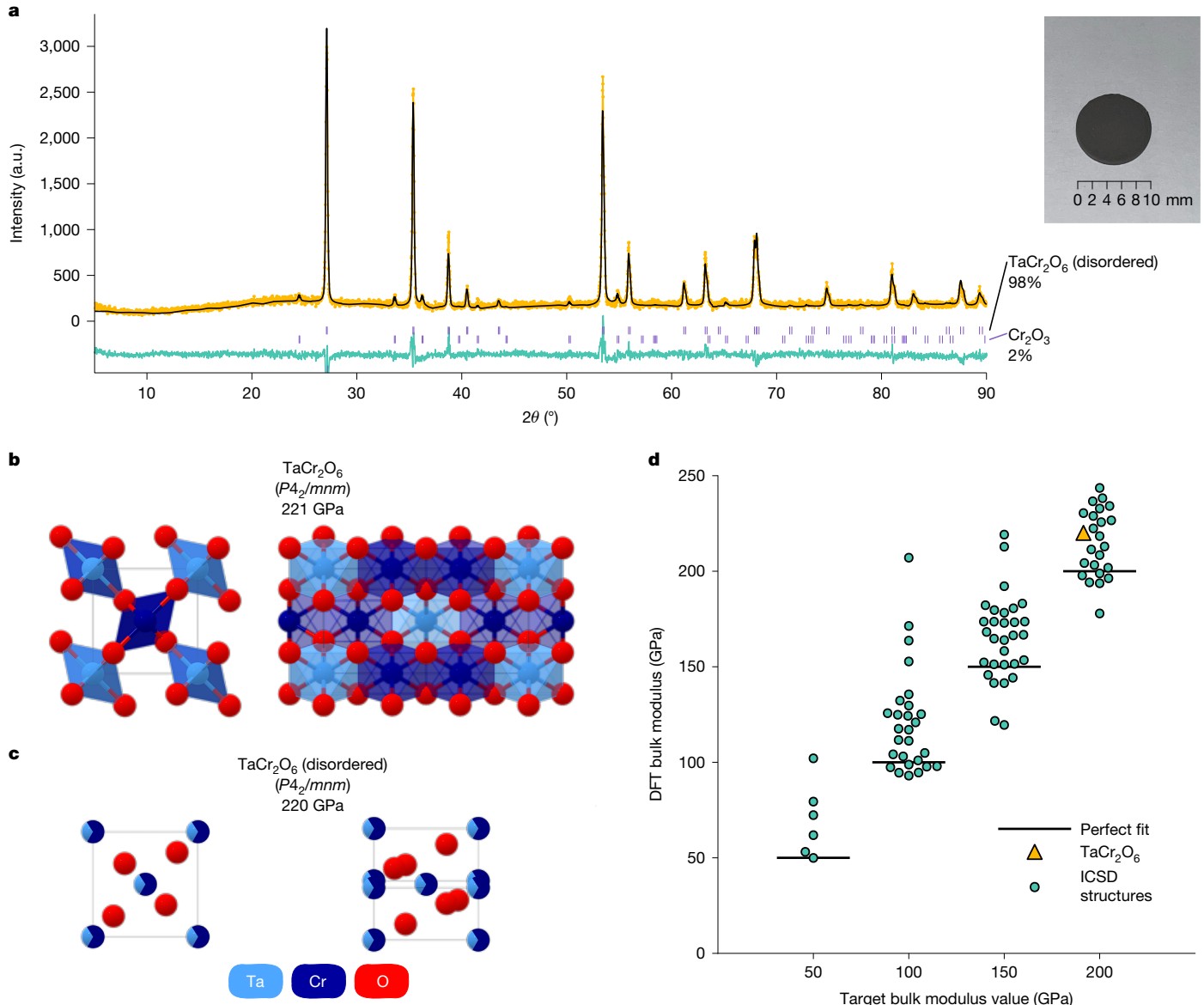

**Fig. 6 | Experimental validation of generated structures. a**, Rietveld refinement for the experimental sample we synthesize, including the measured X-ray diffraction spectra (yellow dots), the theoretical fit (black line) and the difference between the two (teal line). Vertical ticks (purple) highlight the major peaks of $TaCr_2O_6$ and $Cr_2O_3$. Inset: a picture of the sample. **b**, Two views of the $TaCr_2O_6$ structure generated by MatterGen that we use as a synthesis target, along with the reduced formula, space group and DFT bulk modulus value. **c**, Two views of the disordered $TaCr_2O_6$ structure we experimentally synthesize. **d**, DFT bulk modulus values of structures generated by MatterGen that match experimentally verified ICSD structures not present in the training dataset, across four different target bulk modulus values. The yellow triangle indicates the generated structure from **b**. a.u., arbitrary units.

the constraint. We also compare with a screening approach, which scans previously unlabelled materials for promising candidates. In contrast to the previous experiment, we fine-tune MatterGen with labels predicted by a machine learning property predictor—the same used for the screening baseline—when the dataset is not fully labelled. MatterGen finds up to 18 SUN structures with magnetic density above 0.2 Å$^{-3}$ using only 180 DFT property calculations (Fig. 4g). As the dataset is fully labelled, there is no screening baseline available. MatterGen also finds substantially more SUN materials with high bulk modulus than screening (Fig. 4h). Whereas the number of structures found by screening saturates with increasing budget, MatterGen keeps discovering SUN structures at an almost constant rate. Given a budget of 180 DFT property calculations, we find 106 SUN structures (with 95 distinct compositions), more than double the number found with a screening approach (40, 28 distinct compositions). By contrast, there are only two materials in the labelled fine-tuning dataset with these high bulk

modulus values. Note that both MatterGen and screening produce multiple structures per chemical system that are unique according to our definition (Supplementary Information section D.4) but could potentially be alloys with different stoichiometries[40].

## Designing low-supply-chain-risk magnets

Most materials design problems require finding structures satisfying multiple property constraints. Although MatterGen can be fine-tuned for any combination of constraints, here we focus on designing low-supply-chain-risk magnets. Since many existing high-performing permanent magnets contain rare earth elements that pose supply chain risks, there has been increasing interest in discovering rare-earth-free permanent magnets[45]. We simplify this task to finding materials with a high magnetic density of 0.2 Å$^{-3}$ and a low Herfindahl−Hirschman index (HHI) score of 1,250, in which a material with an HHI score below

1,500 is considered to have a low supply chain risk[46] (Supplementary Information section D.9.1). In practice, more properties such as high coercivity, suitable magnetic order and critical temperature need to be satisfied.

In Fig. 5a, we observe that MatterGen generates SUN structures that are narrowly distributed around the target values, despite the labelled fine-tuning data being extremely scarce in that region. Compared with a model that targets only high magnetic density values (single), targeting both properties (joint) shifts the distribution of HHI scores closer towards the desired target value while retaining high magnetic density values. Owing to the lower HHI scores, elements such as cobalt (Co) and gadolinium (Gd) that are often found in magnets with supply chain issues have been almost completely eliminated from the structures generated by the jointly fine-tuned model (Fig. 5b). We show some of these structures in Fig. 5c and analyse them in more detail in Supplementary Information section D.9.2. Finally, we find that MatterGen has rediscovered 67 previously synthesized, disordered structures from ICSD that were unseen during training, many of which are similar to known permanent magnetic materials (Supplementary Information section D.9.3).

## Experimental validation

As a proof of concept, we experimentally synthesize a material designed by MatterGen and show that the experimentally measured property is close to our design target. We generate 8,192 candidates using a model fine-tuned on bulk modulus for each of the four target bulk modulus values: 50 GPa, 100 GPa, 150 GPa and 200 GPa (Supplementary Information section D.10.1). We perform multiple rounds of filtering based on (1) uniqueness and novelty; (2) energy above the hull stability from MatterSim[44] and DFT; (3) phonon stability from MatterSim[44]; and (4) whether the material contains oxygen (Supplementary Information section D.10.3). The filtering narrows the number of candidates down to 75, from which we select four for experimental synthesis after expert inspection. Synthesis was successful for one of the four candidates (Supplementary Information sections D.10.4 and D.10.5). According to the Rietveld refinement analysis, the synthesized material is $TaCr_2O_6$, a compositionally disordered version of the ordered structure predicted by MatterGen (Fig. 6a–c and Supplementary Information section D.10.6). This structure was generated by targeting a bulk modulus value of 200 GPa; we predict a value of 222 GPa using DFT for the ordered $TaCr_2O_6$ structure generated by MatterGen and similar bulk modulus values (219 GPa) for two other ordered approximations corresponding to the same disordered structure (Fig. 6c). We also experimentally measure the Young's modulus of the sample by nanoindentation and estimate its bulk modulus using the DFT-computed Poisson ratio of 0.30. The estimated bulk modulus is up to 169 GPa after four measurements ($158 \pm 11$ GPa), in which the maximum of the four measurements is our best estimate given that the experimental powder sample is likely non-compact (Supplementary Information section D.10.8).

By examining the original 8,192 samples generated for each of the four target values, we find that MatterGen has rediscovered experimentally verified ICSD compounds not present in our training set (Supplementary Information section D.10.2). We identify 101 matches according to our ordered-disordered structure matcher and successfully compute DFT bulk modulus values for 95 of them (Fig. 6d). The DFT-computed values align well with the target values used for conditional generation, with a mean absolute error of 23 GPa and a root mean squared error of 32 GPa.

## Discussion

Generative models are promising for tackling inverse design tasks as they can efficiently explore new structures with desired properties.

However, generating the three-dimensional (3D) structure of stable crystalline materials is challenging because of their periodicity and the interplay between atom types, coordinates and lattice. MatterGen improves on the limitations of previous methods by introducing a joint diffusion process for atom types, coordinates and lattice, which—combined with a vastly larger training dataset—substantially increases the stability, uniqueness and novelty of the generated materials. MatterGen can be fine-tuned to generate SUN structures satisfying target constraints across a wide range of properties, with performance improvements over widely used methods such as MLFF-assisted RSS and substitution, as well as ML-assisted screening. We verified that MatterGen can generate synthesizable structures by experimentally synthesizing a sampled structure and by rediscovering previously synthesized materials that were unseen by the model.

Despite these advances, MatterGen could still be improved in several ways. For example, we observe that the model disproportionately generates structures with P1 symmetry compared with the training data, indicating a tendency for generating less symmetric structures, especially for larger crystals (Supplementary Information section D.2). We propose that further improvements in the denoising process, the backbone architecture and the expansion of the training dataset could enable the model to overcome such issues. We also acknowledge that our evaluations only cover some of the criteria required for real-world applicability, with experimental validation and characterization being the ultimate test[40]. We discuss the challenges in evaluating the quality of crystalline materials from generative models in Supplementary Information section D.2.

We believe that the breadth of the abilities of MatterGen and the quality of generated materials represent an important advance towards creating a universal generative model for materials. Given the enormous impact of generative models in domains such as image generation[47] and protein design[48], we believe that models such as MatterGen will equally transform materials design in the coming years. As such, we are excited about the many directions in which MatterGen could be extended. For instance, MatterGen could be expanded to cover a broader class of materials ranging from catalyst surfaces to metal–organic frameworks, enabling us to tackle challenging problems such as nitrogen fixation[49] and carbon capture[3]. The property constraints can be extended to non-scalar quantities such as the band structure or X-ray diffraction spectrum, which would enable applications ranging from band engineering to the prediction of atomic structures of experimentally measured X-ray diffraction spectra of unknown samples.

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

## Data availability

Alex-MP datasets for training and fine-tuning the MatterGen model are available at GitHub (https://github.com/microsoft/mattergen), along with CIF files for crystal structures presented in the paper, load-depth profiles for nanoindentation measurements, the measured X-ray diffraction profile and the Rietveld refinement for the $TaCr_2O_6$ sample. MP structures (v2022.10.28) are from https://materialsproject.org and Alexandria structures are from https://doi.org/10.24435/materialscloud:m7-50, both under CC BY 4.0 licence. Identifiers of ICSD structures (release 2023.1) used as part of our test set are provided in Supplementary Information; structures are available at https://icsd.products.fiz-karlsruhe.de under a commercial license.

## Code availability

The source code for MatterGen is available at GitHub (https://github.com/microsoft/mattergen).

**Acknowledgements** We thank our colleagues from Microsoft Research AI for Science for their contributions and support, including A. Foong, B. Veeling, Y. Xie, K. Strauss, K. Yan, C. Bodnar, R. van den Berg, F. Noé, M. Segler, E. van der Pol, M. Welling, R. Howard, T.-Y. Liu, B. Kruft and C. Bishop; the Microsoft Azure Quantum team, including C. Chen, L. Talirz and N. Baker, the Materials Project team and Chris Pickard for providing feedback; and the AI on Xbox team for providing part of the computing.

**Author contributions** A.F., M.H., R.P., R.T., T.X., C.Z. and D.Z. conceived the study, implemented the methods, performed computational experiments and wrote the paper. X.F. led the development of the adapter modules. Z.W., C.Y. and W.L. led the experimental synthesis and characterizations. A.S. implemented and ran the symmetry-conditioned generation. J.S. implemented the bandgap workflow. B.N. proposed the task of low-supply-chain risk magnets. Z.L., Y.Z., H.Y., H.H. and J.L. developed the machine learning force field. X.F., A.S., J.C., L.S., J.S., B.N., H.S., S.L., C.-W.H., Z.L., Y.Z., H.Y., H.H. and J.L. helped with implementing the methods, conducting computational experiments and writing the paper. S.U. and R.S. acted as project managers. T.X. and R.T. led the research. C.Z., R.P., D.Z., A.F., M.H., R.T. and T.X. contributed equally; the non-corresponding authors are listed in random order.

**Competing interests** A.F., M.H., R.P., R.T., T.X., C.Z. and D.Z. are inventors of the pending, non-provisional patent application 18/759,208 in the name of Microsoft Technology Licensing, relating to generative models for the computational design of materials. The other authors declare no competing interests.

**Additional information**
**Correspondence and requests for materials** should be addressed to Ryota Tomioka or Tian Xie.
