## [Peer Review File · Nature]

A generative model for inorganic materials design

Corresponding Author: Dr Tian Xie

Version 1:

Reviewer comments:

Referee #1

(Remarks to the Author)

The manuscript 'MatterGen: a generative model for inorganic materials design' proports to address some of the deficiencies of earlier works on the generative models. These include a low success rate in identifying stable crystals and limited ability to impose constraints on properties. The novelty is 'a new diffusion-based generative process that produces crystalline structures by gradually refining atom types, coordinates, and the periodic lattice'. The authors further claim that their approach is 'twice as likely to be novel and stable, and more than 15 times closer to the local energy minimum'. The demonstrated their model on 'structures that have both high magnetic density and a chemical composition with low supply-chain risk.' The manuscript contains ample appendices and supplementary data and citations from the relevant literature. Overall, the paper is intriguing, well written and will certainly grab attention from those in the materials design and discovery community. There are two main parts of the paper that I am concerned about. First of all, I would have liked to have seen more details on how well their model can successfully predict known compounds and their properties. For instance, there are many well-studied ferromagnetic compounds in the mix of elements shown in Fig. 6. How well does their model predict the properties of those known compounds and does it get the space group and site positions and occupancies correct? This goes to my next concern which is the emphasis on unique (being the only one of its kind; unlike anything else) and novel (a new and unusual kind; different from anything seen or known before) which are in this context, are essentially synonyms. While this provides a nice acronym S.U.N., I'm also concerned that the approach results in a very large number of very low symmetry compounds (this is acknowledged on pg. 12). It has been well recognized that the majority of known compounds fall into a small number of space groups (for instance see Journal of Structural Chemistry, Vol. 50, Supplement, pp. S22-S37, 2009 and refences in that paper). This suggest that there is something about certain arrangements of inorganic compounds that are favored over others. While I don't see this as fatal flaw in the manuscript, it does suggest that there is something besides just formation energy relative to the convex Hull is missing. While this topic is beyond the scope of this paper, it does point out a weakness: why choose 14 space groups at random? What would have been more informative to have chosen a favored and non-favored space group in each of the Bavis lattices. Also, all the CIF files in the supplemental data are P1! Am I missing something here?

Specific comments:

Pg. 2 – "MatterGen more than doubles the percentage of generated stable, unique, and novel (S.U.N.) materials, and generates structures that are more than 15 times closer to their ground-truth structures at the DFT local energy minimum." This sentence needs a reference.

Pg. 4 –

"...diffusion takes a symmetric form and approaches a distribution whose mean is a cubic lattice with average atomic density from the training data (Appendix A.7)" Many compounds are far from a cubic average. How might this assumption affect searches for very anisotropic compounds?

"...section, we focus on the ability of MatterGen's base model to generate stable, diverse materials, which we argue is a prerequisite for addressing any inverse materials design task" It was not clear to me how the details of the electronic structures are then related to desired properties?

"...We consider a structure to be stable if its energy per atom after relaxation via DFT is below the 0.1 eV/atom threshold of the reference data set..." How will this capture relative stabilities between compounds within the same phase space, that is on the same convex Hull diagram.

Pg. 6

"Moreover, we benchmark MatterGen against previous generative models for materials and show a significant improvement in performance. We focus on two key metrics:

(1) the percentage of S.U.N. materials among generated samples, measuring the overall success rate of generating promising candidates, and (2) the average RMSD between generated samples and their DFT-relaxed structures, measuring the distance to equilibrium. We also compare to MatterGen-MP, which is a MatterGen model trained only on MP-20, i.e., the same, smaller, dataset used by the other baselines. In Fig. 2(e-f), MatterGen-MP shows a 1.8 times increase in the percentage of S.U.N. structures and a 3.1 times decrease in average RMSD compared with the previous state-of-the-art CDVAE [4]. When comparing MatterGen with MatterGen-MP, we observe a further 1.6 times increase in the percentage of S.U.N. structures and a 5.5 times decrease in RMSD as a result of scaling up the training dataset.” How do you distinguish between generated compounds which are chemically and structurally distinct but closely related. For instance, compounds with differences in site occupancy or small differences in Wyckoff positions but otherwise are the same space groups?

Fig. 3 - Note that matter-gen doesn't do well on the partially or not explored system types. Please comment.

Also note: Code availability statement

MatterGen diffusion model implementation and training script will be made available upon the publication of the manuscript.

Referee #2

(Remarks to the Author)

A: Summary

The paper introduces a deep generative diffusion model for computational inorganic materials generation. Methodologically, it defines a diffusion model jointly over atom positions, atom types and the periodic lattice parameters, incorporating equivariances and invariances in the design where appropriate. It provides strong results and outperforms previous models by a large margin. More specifically, the model can generate stable, unique and novel (SUN) materials with much higher probability than previous works. The model can also be conditioned on target chemistry or symmetry, and be used to generate and design materials with specific with target magnetic, electronic, and mechanical properties. Overall, the model outperforms both previous deep generative models and classical material design approaches (like screening) significantly in the provided benchmark experiments.

B: Originality and significance

I believe the paper is both original and significant. It improves performance over previous generative models for inorganic compound synthesis by a large margin and is maybe the first such model relevant for practical applications, making this potentially highly impactful. The experiments that show conditional generation of materials with important target properties such as target magnetic or electronic structure demonstrate this in particular. The overall method is not complicated: it is "just" a well-designed diffusion model over atom positions, atom types and lattice without a lot of bells and whistles. I think this is great, because it means that the method will be widely adopted both by practitioners and researcher building on top of it. However, I also think the approach is original, as the diffusion model is quite carefully designed in how the score functions are parametrized and such, for instance for the lattice diffusion (Appendix A). I believe MatterGen is, to the best of my knowledge, novel and original in that regard, and this careful diffusion model design is likely what explains its strong performance.

C: Data & methodology: validity of approach, quality of data, quality of presentation

I do not have any concerns with respect to data and methodology. The data and its processing are described in the appendix, and my understanding is that the model is maybe the first such generative model operating at data at this scale, when considering their full Alex-MP-20 dataset. This is interesting and important. With respect to the methodology, I studied the diffusion model details in appendix A in detail and everything seems to be correct and like an appropriate and valid approach to me.

D: Appropriate use of statistics and treatment of uncertainties

Overall, I do not have any major concerns here. To me, the way statistics and evaluations are performed seems to be in line with what is standard practice in the machine learning and generative modeling literature. There is one question about it, though (see 4. in F below).

E: Conclusions: robustness, validity, reliability

Since the model is "just" a big, smartly designed diffusion model without a lot of additional components, and diffusion

models are at this point known to be overall quite reliable and robust, I can believe that the method indeed is quite robust and reliable in practice. Additionally considering the significance and originality, overall, I do support publishing this work in Nature. However, I do have some concerns, questions and suggestions for improvement that I would like to see addressed (see next).

F: Suggested improvements: experiments, data for possible revision

While I am overall positive, I do have suggestions for improvement and some questions, which I would like to see addressed by the authors:

1. Can different adapter modules be combined? This would be interesting and a discussion on this could improve the work.
2. The work considers materials as stable if they are closer than 0.1 eV/atom to a reference dataset after DFT relaxation. Why the 0.1 threshold specifically? Related, does this imply that MatterGen itself, without following DFT relaxation, generally generates somewhat unstable structures and only after DFT relaxation we get the desired outputs? The need for DFT does slow down generation, I believe, and a discussion on why that is and how to address this in future work would be beneficial.
3. One of the main baselines for the method is the "Crystal Diffusion Variational Autoencoder for Periodic Material Generation" (CDVAE) and this is maybe the most related similar method. MatterGen outperforms CDVAE significantly and, looking at CDVAE, it does make sense. However, I believe a critical discussion (in the appendix) of CDVAE would significantly benefit the paper. For instance, the following questions should be discussed: How exactly are MatterGen and CDVAE different, methodologically? Why is it that MatterGen performs so much better? What is the intuition? Aside from CDVAE specifically, typical machine learning papers usually include such discussions in their related works section. Adding a proper discussion of related works and their methodological relations in the appendix of the paper would be very helpful here, too.
4. What is happening in Figure 3d? What is the intuition why MatterGen performs so well for 3 and 5 elements, but not for 4? This dip in performance is not expected. Or is this just statistics, i.e. variance, because using only 9 systems for each case is too less to get reliable results? In that case, the results from this particular experiment are somewhat questionable and not convincing due to high variance. This should be addressed.
5. When the model is fine-tuned to generate compounds with specific properties like space groups, why does it nevertheless generate only <40% of such materials (Fig 4.a). Is it not possible to "overfit" the model to only produce materials from the desired space groups? I can see that generating materials from such specific space groups is generally challenging, but more discussion would be helpful. In other domains, for instance in computer vision, it is usually possible to fine-tune diffusion models to certain target generation.
6. In equation A12 in appendix A.5, the first term seems to explicitly indicate conditioning on the time step "1" in p_{θ} , while the p_{θ} in the last term does not. Is this missing? I would assume the model is generally conditioned on the diffusion time.
7. Is equation A23 in A.6.2 missing a minus sign?
8. Also in A.6.2, the authors describe that they use a truncated sum. This makes sense. However, more details are necessary. How much truncation? How do the results vary for different truncation choices for the sum?
9. I would suggest the authors to add more details to A.7.1; this section is somewhat confusing. The first paragraph discusses the lattice matrix decomposition and the second paragraph how symmetric noise is added. Can the authors please connect this better? Which matrix specifically now gets the noise? How is the decomposition in the first paragraph connected to the symmetric noise addition in the second part? This just is not entirely clear.
10. In B.1, the authors write "When fine-tuning finishes, the score network is able to predict both conditional and unconditional scores." Did the authors (a) train one model that optionally drops out the training labels while fine-tuning the *entire* network include the base model layers? Or (b) did the authors simply fully fine-tune their base model without label dropout and obtain separate conditional and unconditional models? Or (c) did the authors only train the additional adapter module layers during fine-tuning but not the layers of the base model, such that the unconditional model is preserved by simply removing the additional layers? It is a bit confusing how exactly this is done and I would like authors to clarify (this also affects B.2.1 when discussing the null embedding).
11. Will all processed datasets and trained models be publicly released? It seems the authors had to run a lot of data processing and DFT calculate on the datasets (see Appendix C). It would be important to release the processed dataset, so other researchers can directly build on this in future work and also independently validate the method.

G: References

Overall, the most relevant references are cited and appropriate credit is given. However, I think it may be helpful to provide more discussions regarding the relation to CDVAE (see above) and maybe other relevant works and the reasons and intuitions why the proposed model is so much better. This can go into the appendix (with a pointer in the main paper).

H: Clarity and context

Aside from minor issues (see suggested improvements above), the paper was overall easy to follow for me. I think it is clear and mostly provides the relevant context.

Referee #3

(Remarks to the Author)

The authors present a diffusion-based generative model, trained on a large dataset of materials, and used to generate new stable materials with given target properties. There is novelty in the dataset used in this study and in the model architecture, but the comparison to existing models lacks detail, making a quantitative comparison (as presented in the paper) hard to impossible to interpret (details below). The model is applied to multiple materials design tasks, and new materials are predicted computationally. Assuming the novelty and stability definition is reliable (which needs further discussions, details below), interesting candidates are presented, but no validation in terms of experimental synthesis and characterization is provided. Overall, even after major revision, I currently do not see the manuscript as outstanding and game-changing enough to qualify for publication in Nature. The major revision should address my concerns regarding novelty and quantitative evaluation/comparison of the presented method, as well as regarding the definitions of stability/novelty of generated materials.

Stability of materials: The paper uses a criterion of below 0.1 eV/atom as a definition of stability. When a stronger criterion of 0 eV/atom is applied, the amount of generated materials drops substantially (by a factor of 6 when compared to MP's convex hull and by a factor of 25 in the case of Alex-MP-ICSD). This would have a very strong impact on the results shown in Figure 2e, which are also referenced in the abstract ("twice as likely to be novel and stable") and other very prominent positions in the manuscript. The stability definition in the work that the authors compare to [4] is unclear (a threshold of 0.08 eV/atom is mentioned but its use is not further specified). This makes any quantitative comparison meaningless.

Furthermore, as pointed out in Bartel et al. (DOI: 10.1038/s41524-020-00362-y), the prediction of energy above hull values by ML models is unreliable in most cases, as the errors in ML models are more uncorrelated compared to (systematic) DFT errors, making the prediction of energy above hull difficult. Thus, without further evaluation by DFT and validation by experiment, the number of actually stable (and furthermore synthesizable) materials found in this manuscript is questionable.

Novelty of materials: In a reply to the recent Nature paper by Szymanski et al. (DOI: 10.1038/s41586-023-06734-w), Leeman et al. (DOI: 10.26434/chemrxiv-2024-5p9j4) pointed out that novelty of materials is not well-defined, and a definition as used in this manuscript ("We consider a structure to be novel if it is not contained in Alex-MP-ICSD") is certainly not enough.

Novelty of methods: The authors mention several times that a novel diffusion process is introduced. Diffusion processes for the generation of crystal structures were introduced before, e.g. cited in [4]. Method-wise, the novelty here is limited to changes in the architecture of the diffusion model. The main difference is the size of the dataset, as the Alexandria dataset is added to the training. The Alexandria dataset is almost 10 times larger than the MP dataset and thus becomes the largest part of the combined dataset. No comparison of the previous model from [4] on the new dataset is presented, which makes it impossible to differentiate between improvements caused by increasing the size of the dataset and improving the model. I am aware that the new model introduced here has more capabilities than the model presented in [4], but common aspects of the results, mainly presented in Section 2.2, should be compared in more detail to allow reliable conclusions regarding the improvements.

State-of-the-art: The authors compare their results to CDVAE [4], but not to the newer publication DiffCSP [55], which is mentioned in the manuscript but for the first time only on page 8 in Section 2.4 of the Results. [55] reports a substantial improvement over [4].

Reference of unpublished work: "We equip all methods with the MatterSim [52] MLFF, which is used to pre-relax and filter the generated structures by their predicted stability before running more expensive DFT calculations.": [52] is a non-published paper, so it is impossible to judge on the quality and reliability of that work.

Code availability: The authors state that the MatterGen code will be made available upon the publication of the manuscript. Thus, it is not possible to ensure that the code is working and the results are reproducible. The diffusion process on atom types, fractional coordinates, and lattices is described in great detail in the SI. Also, the model architecture and hyperparameters, as well as the datasets are described well. It is likely that this will allow reimplementing of the model, even though it is very likely that not all relevant details can be reproduced without access to the original source code for training and testing of the models.

Figure 2d: Why is the novelty rate dependent on the number of generated samples? Are samples generated repeatedly? How do the authors explain that, if the pool of potential structures is 10^{10} or more, as stated in the introduction? Do you assume that the model suffers from some form of mode collapse, or do you assume that the model is hitting the limit of all possible structures?

(Remarks on code availability)

No code was provided.

Version 2:

Reviewer comments:

Referee #1

(Remarks to the Author)

Overall I found that the revised manuscript to have addressed the major concerns that I had. There were substantial changes made to the manuscript and to the appendices. The author's responses were complete. In particular, there were substantial

changes to appendix D on validating the approach with known compounds, experimental validation and improved clarity on the model and its limitations. The section on ferromagnetic compounds while validating the model also points out a limitation which I think should be noted but not a 'fatal flaw' due to the complexity of magnetic compounds. Specifically, their prediction of the magnetic density of $\text{Ca}_1\text{Mn}_4\text{Fe}_3\text{O}_{12}$, which is experimentally observed to be antiferromagnetic thus 'resulting in a magnetic density that is lower than if it was ferromagnetic'. When calculating the magnetic flux density, how the ferromagnetic ions couple is dependent on their location, ordering on sub-lattices, and presence of anions and/or interstitials which can affect their separations etc. So, when searching for functional materials like permanent magnets (PM), it is important to not only get the crystal structure but the magnetic structure as well. To be a commercially viable PM, the compound also needs to have high coercivity, which is related to its magneto-crystalline anisotropy (MCA). A large MCA however is a necessary but not a sufficient condition for high coercivity, there are extrinsic factors associated with the microstructure that control how much coercivity can be achieved. And of course, this all has to be in a temperature range needed by an application. So, while MatterGen I believe has made some nice inroads to this very complex challenge, there is still a lot that needs to be done. Pointing this out would help put this work in context of the larger challenge.

Referee #2

(Remarks to the Author)

I had provided a detailed and extensive review of the initial version of the manuscript. I have read the authors' reply to this initial review and looked at the revised manuscript. My raised questions and concerns have been addressed in a satisfactory manner, and sufficient modifications have been implemented in the manuscript. I also appreciate the additional evaluations and the experimental synthesis of some materials, which provide further value. The use of statistics seems appropriate and error bars, when shown, are explained in the figure caption. Overall, I support publication of the manuscript.

(Remarks on code availability)

I have not tested the code myself, but the authors have explicitly promised public code, data and model release in their reply. Hence, I am satisfied.

Referee #3

(Remarks to the Author)

Thank you very much for substantially extending the manuscript and carefully replying to all comments of the three reviewers. My main concerns are addressed very well:

- 1) Stability concerns are discussed and explained well
- 2) Additional novelty metrics are introduced
- 3) Additional ablation studies are introduced to disentangle data from model improvements
- 4) Additional baselines (DiffCSP) are introduced
- 5) Minor comments are addressed in a satisfying way
- 6) Code availability: The authors sent the code to the reviewers and will release it after the publication of the manuscript. This needs to be checked by the editorial team.

Overall, these improvements significantly increase the quality of the article. Thus, I support publication in Nature.

(Remarks on code availability)

readme and environment file are present and instructive.

I did not install/run the software but the code seems well-commented and understandable.

If it works, it will be a very valuable resource for the community.

Referee #4

(Remarks to the Author)

1. In TaCr_2O_6 , if the valence of Cr is 3+, that of Ta should be 6+, which is not a table state.

2. Cr_2MoO_6 : MoCr_2O_6 , with trirutile type crystal structure, was synthesized at high pressure and high temperature (Journal of Solid State Chemistry, 1978, 23 [3-4], 315-319). Therefore, it is not surprising that the authors could not synthesize Cr_2MoO_6 at ambient pressure in air.

3. In LaMoO_4 , if the valence of Mo is 6+, that of La should be 2+, which is not a stable state.

4. In Mn_3NiO_6 , the valences of Mn and Ni should be 3+, while Ni with valence of 2+ is stable at high temperature. Therefore, it is reasonable that the authors obtained NiMn_2O_4 with spinel structure, while manganese oxide should be also observed in the XRD pattern.

5. When using high energy ball milling, it is important to state conditions/parameters, such as milling media, materials of vial/balls, diameter of balls, ball-to-powder weight ratio, quantity of water/alcohol used, rotation speed,

Version 3:

Reviewer comments:

Referee #4

(Remarks to the Author)

(1) Regarding TaCr₂O₆, the authors have indicated that Cr is present with both 3+ and 4+ valences simultaneously, which explains the given composition. But, Cr⁴⁺ is not very commonly observed.

(2) Noting that the rest three samples are not successfully synthesized, why did not the authors try other compositions?

Response to referees for manuscript “MatterGen: a generative model for inorganic materials design”

We thank all referees for their constructive feedback and insightful comments. We are glad you found our manuscript to be “intriguing” and “well written” (R1)—concluding that it “will certainly grab attention from those in the materials design and discovery community” (R1)—and found our generative model to be “maybe the first such model relevant for practical applications, making this potentially highly impactful” (R2). We also appreciate that you highlighted our contributions, including “novelty in the dataset [...] and in the model architecture” (R3). Below, we briefly summarize the major revisions of the manuscript. Thereafter, we provide detailed responses to each referee’s comments.

Validation through experimental measurements and DFT We have successfully synthesized TaCr_2O_6 , a material that was conditionally generated by MatterGen with a bulk modulus target of 200 GPa (Section 2.6). The synthesized material is novel and matches the crystal structure of the generated material, but it has cation disorder between the Ta and Cr sites. The experimentally measured bulk modulus is estimated to be 158 ± 11 GPa, and the DFT computed bulk modulus is 220 ± 1 GPa, both close to our target value. This result provides strong evidence that our model is indeed able to design novel, synthesizable materials with desired target properties. In addition, we have validated the efficacy of our approach by rediscovering previously synthesized structures from the ICSD database that were not seen by the model during training. In particular, we find up to 2,500 ordered (and up to 8,250 disordered) structures when sampling unconditionally (Appendix D.5.4), 67 disordered structures when searching for materials with a high magnetic density and low HHI score (Appendix D.9.3)—many of which are similar to known permanent magnetic materials—and 101 structures when generating materials with target bulk modulus values (Appendix D.10.2). Finally, we have revised the manuscript to further clarify that all computational evaluations are based on DFT (Appendix D.3).

Comparison with other methods We have substantially expanded the comparison of MatterGen with previous SOTA materials discovery approaches. In particular, we show that MatterGen outperforms DiffCSP [5] in the unconditional performance benchmark (Fig. 2(e-f)). Furthermore, we have included an additional qualitative (Appendix A.11) and quantitative (Appendix D.5.2) analysis, including comparisons with CDVAE and DiffCSP trained on different datasets. Lastly, we have performed an extensive ablation study of our model (Appendix A.10) that provides further insights into where the performance improvements of MatterGen are coming from.

Definition of novelty and stability We have updated our definition of novelty to consider compositional disorder effects (Appendix D.4), and clarified our definitions of stability, uniqueness and novelty in the manuscript (Appendix D.3). Concretely, we have developed a new disordered structure matching algorithm inspired by the well-known Hume-Rothery rules [2] (Appendix D.4) and significantly expanded our reference dataset by including 117,700 compositionally disordered structures from ICSD (Appendix C.1). Moreover, we have validated the efficacy of the disordered structure matcher in a benchmark study. Consequently, we have revised all qualitative and quantitative analyses in the manuscript according to the revised definition. We believe the disordered structure matcher will significantly raise the bar of evaluating computationally designed materials [6]. Therefore, we aim to make the algorithm available to the community by open-sourcing it via the `pymatgen` package.

Code and data release We share a preliminary version of the code and data release, which can be used for the purpose of reviewing this work. The code includes the MatterGen model implementation as well as training and evaluation scripts. The data includes the training and reference datasets as well as CIF files of all structures highlighted in the manuscript. We are ready to release the code and data upon publication of the manuscript.

1 R1: rare earth free magnets discovery

General comments *The manuscript ‘MatterGen: a generative model for inorganic materials design’ proports to address some of the deficiencies of earlier works on the generative models. These include a low success rate in*

identifying stable crystals and limited ability to impose constraints on properties. The novelty is ‘a new diffusion-based generative process that produces crystalline structures by gradually refining atom types, coordinates, and the periodic lattice’. The authors further claim that their approach is ‘twice as likely to be novel and stable, and more than 15 times closer to the local energy minimum’. They demonstrated their model on ‘structures that have both high magnetic density and a chemical composition with low supply-chain risk.’ The manuscript contains ample appendices and supplementary data and citations from the relevant literature. Overall, the paper is intriguing, well written and will certainly grab attention from those in the materials design and discovery community.

Thank you for carefully reviewing our manuscript and for your insightful comments and questions. We have significantly revised our manuscript based on your feedback, which has helped us to further improve the quality of our work. Please find a summary of the most important changes and our point-to-point response below.

- **Rediscovery of known compounds:** We have performed an in-depth analysis about how well our model is able to rediscover ferromagnetic compounds (Appendix D.9.3). In addition, we have added experiments to demonstrate the model’s ability to rediscover known compounds unconditionally (Appendix D.5.4), and when conditioned on bulk modulus (Appendix D.10.2).
- **Definition of stability, uniqueness and novelty:** We have clarified the definitions of stability, uniqueness and novelty. This includes the difference between uniqueness and novelty, the role of symmetry, how closely related structures are distinguished, and how relative stability between compounds is assessed.
- **Improved clarity and analysis:** We have improved clarity about the model and its limitations. Additionally, we have improved the analysis of symmetry-constrained generation (Appendix D.7).

Q1.1 *I would have liked to have seen more details on how well their model can successfully predict known compounds and their properties. For instance, there are many well-studied ferromagnetic compounds in the mix of elements shown in Fig. 6. How well does their model predict the properties of those known compounds and does it get the space group and site positions and occupancies correct?*

Thank you for the suggestion. We have added a detailed analysis of how well our model can rediscover known ferromagnetic compounds that were not included in the training set of the model (Appendix D.9.3). Using our updated structure matcher that accounts for compositional disorder effects (Appendix D.4), we find 18 unique samples from the conditional model in Section 2.5 that match 67 compositionally disordered ICSD structures, all of which have been experimentally verified previously (Table D8). We have searched the scientific literature for experimental measurements of the magnetic density for the 67 re-discovered ICSD compounds and we include a non-exhaustive list of compounds that have a similar formula and space group along with the measured magnetic moment. All but one of the experimental measurements we found report ferromagnetic ordering of the compound, which is consistent with our calculations that assume ferromagnetic ordering. Likewise, we highlight that the DFT-computed magnetic densities for these structures closely match the conditional generation target. The only exception is given by the $\text{Ca}_1\text{Mn}_4\text{Fe}_3\text{O}_{12}$ compound, which is experimentally observed to be antiferromagnetic and therefore has a magnetic density that is lower than if it was ferromagnetic. Finally, we find that the structures generated by MatterGen match the space groups of the experimentally known disordered magnetic structures quite well. All $\text{Fe}_{1-x}\text{Ni}_x$ MatterGen compounds shown for example, have an equivalent disordered-structure space group (see the table caption for a detailed explanation) of $\text{Im}\bar{3}\text{m}$. This matches the space group for ferromagnetic Fe referenced in the literature.

Q1.2 *This goes to my next concern which is the emphasis on unique (being the only one of its kind; unlike anything else) and novel (a new and unusual kind; different from anything seen or known before) which are in this context, are essentially synonyms. I’m also concerned that the approach results in a very large number of very low symmetry compounds (this is acknowledged on pg. 12). It has been well recognized that the majority of known compounds fall into a small number of space groups (for instance see Journal of Structural Chemistry, Vol. 50, Supplement, pp. S22-S37, 2009 and references in that paper). This suggests that there is something about certain arrangements of inorganic compounds that are favored over others. While I don’t see this as fatal flaw in the manuscript, it does suggest that there is something besides just formation energy relative to the convex Hull is missing.*

We agree that the widely adopted approach to evaluate stability with respect to the 0K energy above the convex hull may result in low symmetry materials being predicted to be stable, and our definitions of unique and novel do not necessarily filter out such structures. We provide a detailed discussion about this and other limitations in Appendix D.2. Developing computationally tractable metrics that better capture synthesizability is still an active area of research. By open-sourcing our model, we enable the community to re-evaluate our model’s performance on future synthesizability metrics as soon as they become available.

In addition, we would like to clarify the difference between uniqueness and novelty. Uniqueness refers to whether a particular structure we generate is distinct from all other structures we generate. As mentioned in the manuscript, we use this as a proxy for measuring the diversity of the generated structures. In contrast, novelty measures whether a particular structure we generate is distinct from all other previously known structures (in our reference dataset). For example, when asked to generate N structures, a model might generate a previously unknown structure N times, in which case novelty would be high but uniqueness would be low. A practically useful model should be able to generate both unique/diverse and novel structures. We have revised the manuscript to make this more clear (Appendix D.3).

Q1.3 [...] why choose 14 space groups at random? What would have been more informative to have chosen a favored and non-favored space group in each of the Bravais lattices.

Thank you for your suggestion. We have revised the entire symmetry section (now in Appendix D.7 due to space constraints) to make the results more informative. In Fig. D9 we investigate how well our model can recover the distribution of space groups in the training data, spanning a wide range of symmetries across all Bravais lattices. In addition, we have updated the original plot (now Fig. D8) by differentiating between the fraction of structures that (1) satisfy the space group constraint, (2) are additionally stable, and (3) are additionally unique and novel. Finally, we have included a more detailed analysis of the model performance based on the more fine-grained results, which together with the qualitative analysis of generated structures (Appendix D.7.2) provide a comprehensive overview of the model’s capabilities.

Q1.4 [...] all the CIF files in the supplemental data are P1! Am I missing something here?

Thank you for pointing this out. We have corrected the space groups in the CIF files.

Q1.5 Pg. 2 – “MatterGen more than doubles the percentage of generated stable, unique, and novel (S.U.N.) materials, and generates structures that are more than 15 times closer to their ground-truth structures at the DFT local energy minimum.” This sentence needs a reference.

Thank you for pointing this out. We have added a reference.

Q1.6 “...diffusion takes a symmetric form and approaches a distribution whose mean is a cubic lattice with average atomic density from the training data (Appendix A.7)” Many compounds are far from a cubic average. How might this assumption affect searches for very anisotropic compounds?

We are not making any assumption about the shape of lattice matrices in the generated samples, and therefore do not bias the search away from anisotropic compounds. To make this more clear, consider the equation for the reverse step in the lattice diffusion:

$$p(\mathbf{L}_{t-1}|\mathbf{L}_t, \mathbf{A}_t, \mathbf{X}_t) = \mathcal{N}\left(\frac{1}{\sqrt{1-\beta_t}}(\mathbf{L}_t + \beta_t\sigma(N)^2 s_{\mathbf{L},\theta}(\mathbf{L}_t, \mathbf{A}_t, \mathbf{X}_t, t)) + (1 - \frac{1}{\beta_t})\mu(N)\mathbf{I}, \beta_t\sigma(N)^2\mathbf{I}\right), \quad (1)$$

where $(1 - \frac{1}{\beta_t}) < 0$. Hence, the bias towards cubic lattices (i.e., adding a fraction of $\mu(N)\mathbf{I}$ to the noisy mean) from the forward diffusion gets ‘subtracted’ gradually during the reverse diffusion process and is fully eliminated after denoising, so that the denoising process does not have any particular bias towards cubic lattices.

The reason why we set the mean of the prior lattice distribution to be a cube with fixed atomic density (Appendix A.7.2) is only to simplify the learning process. If we instead choose a $\mathcal{N}(0, 1)$ prior for each lattice matrix element, the resulting random lattices will have very high atomic density (i.e., small volume) on average. Since we convert the noisy fractional coordinates to Cartesian coordinates before feeding them into the model, the backbone will have to deal with vastly different scales of interatomic distances over the course of diffusion/denoising: for clean data at $t = 0$, atomic distances are on the order of a few Å, whereas at $t \rightarrow 1$ many of them are of order 10^{-3} Å. This poses challenges for model learning, whereas choosing a cube with fixed atomic density as the mean of the prior distribution leads to similarly-scaled interatomic distances across the diffusion and denoising trajectories. See also the newly added ablation study (Appendix A.10), where we show that using a standard diffusion process for the lattice leads to no samples being successfully generated from the model.

Q1.7 “...section, we focus on the ability of MatterGen’s base model to generate stable, diverse materials, which we argue is a prerequisite for addressing any inverse materials design task” It was not clear to me how the details of the electronic structures are then related to desired properties?

The main message of this sentence is that we first need MatterGen to generate stable, near-equilibrium structures before addressing property-guided inverse design tasks. For most properties like bulk modulus and

band gap, it is most meaningful to compute these properties using the equilibrium structure. Therefore, if MatterGen cannot generate stable material structures, it is difficult to imagine it will perform well in property-guided inverse design. By fine-tuning the base model, MatterGen learns the relationship between structure and properties from the labeled fine-tuning dataset. The model does not explicitly represent electronic structure; instead, it learns from the atomic structure of nuclei using atomic coordinates (Appendix A.8).

Q1.8 *“... We consider a structure to be stable if its energy per atom after relaxation via DFT is below the 0.1 eV/atom threshold of the reference data set...” How will this capture relative stabilities between compounds within the same phase space, that is on the same convex Hull diagram.*

If we generate multiple compounds within the same convex hull diagram whose energies per atom are below the 0.1 eV/atom threshold, we consider all of them to be stable, irrespective of whether all structures would still be below the threshold when considering the combined convex hull. However, note that for most experiments the probability of generating two structures that are on the same convex hull diagram is relatively low. We have updated the manuscript to clarify this (Appendix D.3). The only exception where we indeed consider the combined convex hull is in Section 2.3, where we generate materials given a target chemical system. That is because we explicitly define the task as finding the most stable structures in a given chemical system.

Q1.9 *Pg. 6 How do you distinguish between generated compounds which are chemically and structurally distinct but closely related. For instance, compounds with differences in site occupancy or small differences in Wyckoff positions but otherwise are the same space groups?*

We previously considered two structures to be the same if they match according to the pymatgen STRUCTUREMATCHER. To account for compositional disorder effects, we have developed a new structure matching algorithm to assess whether a pair of structures can be identified as ordered approximations of the same underlying disordered structure or, similarly, whether an ordered structure can be identified as an ordered approximation of a known disordered structure (Appendix D.4). Under both the pymatgen structure matcher and our novel disordered structure matcher, two compounds with small differences in Wyckoff positions are considered the same if they are within given tolerances, even if they have different space groups according to the pymatgen SPACEGROUPANALYZER. We have clarified this in Appendix D.3.

Q1.10 *Fig. 3 - Note that matter-gen doesn't do well on the partially or not explored system types. Please comment.*

MatterGen outperforms the baseline methods in all chemical system exploration scenarios when considering the S.U.N. metric. When looking at the ‘Structures on the combined hull’ metric, MatterGen performs slightly better than the baselines on ‘partially explored’ systems, and worse than the baselines on ‘not explored’ systems. The performance of MatterGen in the latter case could be caused by the more uncommon element combinations found (by design) in the ‘not explored’ systems, forcing the model to generate structures that are outside the training set distribution. We expect the model performance to improve given more data in these regions (still assuming the particular chemical system is unseen).

Alternatively, the variation in performance could be caused by the relatively small number of chemical systems considered here (27). While additional calculations could be run to expand the number of chemical systems, these would require a very substantial computational effort, e.g., for each chemical system 600,000 structures are generated and relaxed for the random structure search baseline.

Q1.11 *Also note: Code availability statement - MatterGen diffusion model implementation and training script will be made available upon the publication of the manuscript*

As mentioned in the summary above, the MatterGen diffusion model implementation and training script are part of the preliminary code release. We are ready to release the code upon publication of the manuscript.

2 R2: diffusion models

General comments (summary) *The paper introduces a deep generative diffusion model for computational inorganic materials generation. Methodologically, it defines a diffusion model jointly over atom positions, atom types and the periodic lattice parameters, incorporating equivariances and invariances in the design where appropriate. It provides strong results and outperforms previous models by a large margin. More specifically, the model can generate stable, unique and novel (SUN) materials with much higher probability than previous works. The model can also be conditioned on target chemistry or symmetry, and be used to generate and design materials with*

specific with target magnetic, electronic, and mechanical properties. Overall, the model outperforms both previous deep generative models and classical material design approaches (like screening) significantly in the provided benchmark experiments.

Thank you for carefully reading our manuscript and for a great summary of our contributions.

General comments (originality and significance) *I believe the paper is both original and significant. It improves performance over previous generative models for inorganic compound synthesis by a large margin and is maybe the first such model relevant for practical applications, making this potentially highly impactful. The experiments that show conditional generation of materials with important target properties such as target magnetic or electronic structure demonstrate this in particular. The overall method is not complicated: it is "just" a well-designed diffusion model over atom positions, atom types and lattice without a lot of bells and whistles. I think this is great, because it means that the method will be widely adopted both by practitioners and researcher building on top of it. However, I also think the approach is original, as the diffusion model is quite carefully designed in how the score functions are parametrized and such, for instance for the lattice diffusion (Appendix A). I believe MatterGen is, to the best of my knowledge, novel and original in that regard, and this careful diffusion model design is likely what explains its strong performance.*

We appreciate that you highlighted the originality and significance of our work.

General comments (data & methodology) *I do not have any concerns with respect to data and methodology. The data and its processing are described in the appendix, and my understanding is that the model is maybe the first such generative model operating at data at this scale, when considering their full Alex-MP-20 dataset. This is interesting and important. With respect to the methodology, I studied the diffusion model details in appendix A in detail and everything seems to be correct and like an appropriate and valid approach to me.*

Thank you for studying the diffusion model details in Appendix A in detail. We have added new ablation studies and more comparisons to baselines in our revised manuscript.

General comments (appropriate use of statistics and treatment of uncertainties) *Overall, I do not have any major concerns here. To me, the way statistics and evaluations are performed seems to be in line with what is standard practice in the machine learning and generative modeling literature. There is one question about it, though (see 4. in F below).*

Thank you for your comments. We have addressed your question in our point-to-point response below.

General comments (conclusions) *Since the model is "just" a big, smartly designed diffusion model without a lot of additional components, and diffusion models are at this point known to be overall quite reliable and robust, I can believe that the method indeed is quite robust and reliable in practice. Additionally considering the significance and originality, overall, I do support publishing this work in Nature. However, I do have some concerns, questions and suggestions for improvement that I would like to see addressed (see next).*

Thank you for supporting the publication of our paper in Nature. We appreciate your detailed feedback and helpful suggestions for further improving the manuscript. In the following, we briefly summarize the main changes we have made based on your feedback and provide point-to-point to your comments below.

- **Discussion of related work:** We have added a discussion about the key differences between MatterGen and related work (Appendix A.11). Moreover, we have added an ablation study (Appendix A.10) that explains where the performance gains of MatterGen are coming from. Finally, we have expanded the quantitative comparison of MatterGen with other baselines (Appendix D.5.2).
- **Improved clarity:** We have revised the manuscript to improve clarity and provide further details, e.g., about the model, DFT relaxation, and experimental results.
- **Code and data release:** We are happy to share a preliminary version of the code and data release, which will enable other researchers to build upon this work and independently validate the method.

Q2.1 *Can different adapter modules be combined? This would be interesting and a discussion on this could improve the work.*

Yes, in principle this is possible. During fine-tuning our model already learns the joint conditional distribution $p(M|c_1, c_2, \dots)$ for different properties c_i . If we further assume these properties are conditionally independent, we could compose multiple single-property conditional models together by following ideas from Liu et al. [7]. We

agree this would be an interesting improvement that we leave for future work. We have updated Appendix B.1 accordingly.

Q2.2 *The work considers materials as stable if they are closer than 0.1 eV/atom to a reference dataset after DFT relaxation. Why the 0.1 threshold specifically? Related, does this imply that MatterGen itself, without following DFT relaxation, generally generates somewhat unstable structures and only after DFT relaxation we get the desired outputs? The need for DFT does slow down generation, I believe, and a discussion on why that is and how to address this in future work would be beneficial.*

We choose a threshold of 0.1eV/atom because Aykol et al. [1] have shown that the amorphous limit for meta-stability is between 0.05 eV/atom to 0.9 eV/atom depending on the chemical system. Our threshold of 0.1 eV/atom represents a trade-off between including some potentially synthesizable metastable materials while not including too many unsynthesizable materials. Chen and Ong [3] consider this threshold for the same reason.

As shown in Fig. 2(c), the majority of structures generated by MatterGen are very close to the DFT local energy minimum (the average RMSD is only around 0.02 Å), implying that the structures are good enough for most practical use cases without DFT relaxation. We only perform DFT relaxation to compute energies and properties for the purpose of evaluation. This is common practice because many properties are only defined w.r.t. the fully relaxed structure (e.g., bulk modulus). Finally, note that it is possible to pre-relax and filter structures using energies predicted by a fast and accurate machine learning force field before DFT. Indeed, we do this for some of the experiments in the manuscript. This significantly reduces the computational cost of DFT relaxation. We have added Appendix D.3 to make these points more clear.

Q2.3 *One of the main baselines for the method is CDVAE and this is maybe the most related similar method. MatterGen outperforms CDVAE significantly and, looking at CDVAE, it does make sense. However, I believe a critical discussion (in the appendix) of CDVAE would significantly benefit the paper. For instance, the following questions should be discussed: How exactly are MatterGen and CDVAE different, methodologically? Why is it that MatterGen performs so much better? What is the intuition? Aside from CDVAE specifically, typical machine learning papers usually include such discussions in their related works section. Adding a proper discussion of related works and their methodological relations in the appendix of the paper would be very helpful here, too.*

Thank you for the suggestion. We have added a discussion about the differences between MatterGen and CDVAE, and which changes we attribute to the improved performance of MatterGen (Appendix A.11). In summary, the key factors for improved performance of MatterGen over CDVAE are: (i) using the score of the wrapped normal distribution as opposed to using a heuristic; (ii) the ability to update the lattice during the generation process as opposed to relying on a potentially flawed initial guess; and (iii) the ‘pure’ diffusion model framework as opposed to a hybrid VAE and diffusion framework, preventing a mismatch between the standard normal prior and the latent variables \mathbf{z} .

We have additionally added a comparison to DiffCSP (Appendix A.11) and report results on unconditional generation in the updated Fig. 2(a-b) in the manuscript. We also train DiffCSP on our large Alex-MP dataset (Table D4), where MatterGen outperforms it by more than 5x on RMSD and 10 % on the fraction of S.U.N. structures. Finally, we have added an ablation study (Appendix A.10) that demonstrates how the individual model innovations improve results, e.g., replacing CDVAE’s heuristic coordinate score with the wrapped normal score increases the S.U.N. percentage by more than 4x and reduces RMSD by more than 10x.

Q2.4 *What is happening in Figure 3d? What is the intuition why MatterGen performs so well for 3 and 5 elements, but not for 4? This dip in performance is not expected. Or is this just statistics, i.e. variance, because using only 9 systems for each case is too less to get reliable results? In that case, the results from this particular experiment are somewhat questionable and not convincing due to high variance. This should be addressed.*

We attribute this result to the variance of this experiment. The very strict criterion of a structure being on the combined hull leads to rare outcomes, for which the variance will inherently be large. Unfortunately, scaling this experiment to more chemical systems requires substantial resources since, in the pursuit of a fair comparison, RSS and substitution were run on a very large number of samples (600K structures generated and relaxed for RSS on each chemical system). Nevertheless, we argue that the overall observations and conclusions from Fig. 3(c-d) remain valid even with one outlier result. Moreover, we refer to Fig. 3(a-b) for results with less variance as the S.U.N. criterion is less strict. Here, MatterGen outperforms all baselines in all settings.

Q2.5 *When the model is fine-tuned to generate compounds with specific properties like space groups, why does it nevertheless generate only < 40% of such materials (Fig 4.a). Is it not possible to “overfit” the model to only produce materials from the desired space groups? I can see that generating materials from such specific space*

groups is generally challenging, but more discussion would be helpful. In other domains, for instance in computer vision, it is usually possible to fine-tune diffusion models to certain target generation.

We agree that generating materials from a specific space group is challenging. However, there appears to be a small misunderstanding with respect to Fig. 4(a) (now Fig. D8(a) in the Appendix): the bars denote the fraction of generated structures which are S.U.N. *and* are from the desired target space groups. For these results, the raw hit rate of the target space group (i.e., ignoring S.U.N.) is 54%. We have revised Fig. D8(a) to improve clarity, and refer to Fig. D9 for more details.

Further, note that it may not be possible to always overfit because the space group of a structure might be sensible to small variations in the atomic positions. This is different from image generation where small differences in pixel space do not change the class/concept of the output. In other cases like F43m, it is difficult to find novel materials because it is a high symmetry space group that is already well explored in the training data. The updated Fig. D8(a)—which now separates S.U.N. and the symmetry hit rate—makes this more clear.

Q2.6 *In equation A12 in appendix A.5, the first term seems to explicitly indicate conditioning on the time step "1" in p_θ , while the p_θ in the last term does not. Is this missing? I would assume the model is generally conditioned on the diffusion time.*

You are right. Thank you for pointing out the typo. We have corrected the equation accordingly.

Q2.7 *Is equation A23 in A.6.2 missing a minus sign?*

Yes, thank you for spotting the typo. We have added the missing minus sign in the updated manuscript.

Q2.8 *Also in A.6.2, the authors describe that they use a truncated sum. This makes sense. However, more details are necessary. How much truncation? How do the results vary for different truncation choices for the sum?*

We truncate all terms with an offset vector \mathbf{k} whose infinity norm is larger than a cutoff value, which we choose to be 13. This results in $(2 * 13 + 1)^3 = 19,683$ terms in the sum. We have added a figure to Appendix A.6.2 that shows the approximation error for different cutoff values, which highlights that the approximation error is negligible given our cutoff.

Q2.9 *I would suggest the authors to add more details to A.7.1; this section is somewhat confusing. The first paragraph discusses the lattice matrix decomposition and the second paragraph how symmetric noise is added. Can the authors please connect this better? Which matrix specifically now gets the noise? How is the decomposition in the first paragraph connected to the symmetric noise addition in the second part? This just is not entirely clear.*

Thank you for the suggestion. We have revised Appendix A.7.1 to improve clarity. In summary, we restrict our entire forward and reverse lattice diffusion to symmetric matrices, effectively eliminating the rotational degrees of freedom. That is, for an input data point with a lattice matrix with arbitrary rotation we first compute the symmetric lattice via Eq. A27 and only use this symmetric matrix going forward. To ensure that the lattice matrix remains symmetric during the diffusion process, we only add symmetric noise to the lattice matrix.

Q2.10 *In B.1, the authors write "When fine-tuning finishes, the score network is able to predict both conditional and unconditional scores." Did the authors (a) train one model that optionally drops out the training labels while fine-tuning the *entire* network include the base model layers? Or (b) did the authors simply fully fine-tune their base model without label dropout and obtain separate conditional and unconditional models? Or (c) did the authors only train the additional adapter module layers during fine-tuning but not the layers of the base model, such that the unconditional model is preserved by simply removing the additional layers? It is a bit confusing how exactly this is done and I would like authors to clarify (this also affects B.2.1 when discussing the null embedding).*

Option (a) is correct: While at initialization we ensure that the model's unconditional outputs are not changed by adding the adapter and mixin layers, during the course of fine-tuning we update all model parameters. We have revised Appendix B.1 for improved clarity.

Q2.11 *Will all processed datasets and trained models be publicly released? It seems the authors had to run a lot of data processing and DFT calculate on the datasets (see Appendix C). It would be important to release the processed dataset, so other researchers can directly build on this in future work and also independently validate the method.*

Yes, we will publicly release the processed datasets and the model. Please take a look at the attached data and code for more details.

3 R3: AI materials discovery

General comments *The authors present a diffusion-based generative model, trained on a large dataset of materials, and used to generate new stable materials with given target properties. There is novelty in the dataset used in this study and in the model architecture, but the comparison to existing models lacks detail, making a quantitative comparison (as presented in the paper) hard to impossible to interpret (details below). The model is applied to multiple materials design tasks, and new materials are predicted computationally. Assuming the novelty and stability definition is reliable (which needs further discussions, details below), interesting candidates are presented, but no validation in terms of experimental synthesis and characterization is provided. Overall, even after major revision, I currently do not see the manuscript as outstanding and game-changing enough to qualify for publication in Nature. The major revision should address my concerns regarding novelty and quantitative evaluation/comparison of the presented method, as well as regarding the definitions of stability/novelty of generated materials.*

Thank you for carefully reviewing our work and for your constructive comments. MatterGen represents a paradigm shift from screening-based towards generative-model-based materials design. For the first time, MatterGen shows that we can guide the exploration of new materials using a broad range of conditions, including chemistry, symmetry, and properties. The exploration of the enormous materials space via a generative model is significantly more efficient than the brute-force screening of millions of material candidates.

We agree that one major limitation of our work has been the lack of experimental synthesis. To address this, we have worked with an experimental laboratory and successfully synthesized a material generated by MatterGen. This material, TaCr_2O_6 , was conditionally generated by MatterGen with a bulk modulus target value of 200 GPa (Section 2.6). The synthesized material is novel and matches the crystal structure of the generated material, but it has cation disorder between the Ta and Cr sites. The experimentally measured bulk modulus is estimated to be 158 ± 11 GPa, while the DFT-computed bulk modulus is 220 ± 1 GPa, both close to our conditional target.

We have also addressed all other concerns that were raised. Below is a summary of the most important changes based on your feedback, followed by point-to-point responses.

- **Novelty and stability of materials:** We have substantially improved our definition of novelty, e.g., by leveraging our newly developed and benchmarked disordered structure matcher (Appendix D.3), and by further expanding our reference dataset (Appendix C.1). In addition, we have clarified our definition of stability. Importantly, we have clarified that all our evaluations use energy values computed by DFT.
- **Quantitative comparison and novelty of method:** We have added and/or expanded comparisons with state-of-the-art methods such as CDVAE and DiffCSP (Section 2.2, Appendix D.5.2). Moreover, we have further highlighted the novelty of our method (Appendix A.11). Finally, we have added an ablation study of our model to demonstrate the effectiveness of our model innovations (Appendix A.10).
- **Experimental validation:** We have experimentally synthesized and characterized a material that was conditionally sampled by MatterGen, demonstrating that our model is able to generate novel, synthesizable materials (Section 2.6, Appendix D.10).

Q3.1 *Stability of materials: The paper uses a criterion of below 0.1 eV/atom as a definition of stability. When a stronger criterion of 0 eV/atom is applied, the amount of generated materials drops substantially (by a factor of 6 when compared to MP’s convex hull and by a factor of 25 in the case of Alex-MP-ICSD). This would have a very strong impact on the results shown in Figure 2e, which are also referenced in the abstract (“twice as likely to be novel and stable”) and other very prominent positions in the manuscript. The stability definition in the work that the authors compare to [4] is unclear (a threshold of 0.08 eV/atom is mentioned but its use is not further specified). This makes any quantitative comparison meaningless.*

We choose 0.1 eV/atom because Aykol et al. [1] have shown that the amorphous limit for meta-stability is between 0.05 eV/atom to 0.9 eV/atom depending on the chemical system. Our threshold of 0.1 eV/atom ensures that enough metastable materials are covered without including too many unsynthesizable materials. Chen and Ong [3] consider this threshold for the same reason. Also note that our experimentally synthesized structure is metastable at 0.024 eV/atom above the hull and would have been erroneously excluded at a 0 eV/atom cutoff.

While our results are sensitive to the chosen stability threshold, we identify the mismatch between the training data and the evaluation target as one of the the main causes for this. Since the model will approximate the training distribution, it is important that the definition of stability is consistent between training and evaluation. Our training set contains 8.0 % of structures stable at 0 eV/atom, while the rate of structures in MP20 that are stable at 0 eV/atom is more than five times higher (44.1 %). Thus, MatterGen generating stable structures (0 eV/atom) at a rate of around 3 % is comparable to our training data and therefore an expected outcome.

To emphasize the importance of aligning the model with the intended outcome (e.g., generating stable structures at 0 or 0.1 eV/atom), we note that it is possible to further improve the percentage of S.U.N. structures at 0 eV/atom by explicitly conditioning on energy above the hull equals 0 eV/atom. In Appendix D.6.3 and the newly added Table D7, we show that MatterGen improves the percentage of S.U.N. structures at 0 eV/atom across chemical systems by jointly conditioning on chemical system and energy above the hull.

Finally, note that the CDVAE paper did not report energy above hull for any of their results. Their 0.08 eV/atom threshold was only used to construct the training data with respect to the Materials Project convex hull at the time of publication. To ensure meaningful comparisons with CDVAE, G-SchNet, P-G-SchNet and FTCP we evaluated publicly released structures with the same DFT pipeline we use throughout this work, and evaluate stability with the same criterion and threshold of 0.1 eV/atom above the Alex-MP-ICSD convex hull.

Q3.2 [...] as pointed out in Bartel et al. (DOI: 10.1038/s41524-020-00362-y), the prediction of energy above hull values by ML models is unreliable in most cases, as the errors in ML models are more uncorrelated compared to (systematic) DFT errors, making the prediction of energy above hull difficult. Thus, without further evaluation by DFT and validation by experiment, the number of actually stable (and furthermore synthesizable) materials found in this manuscript is questionable.

We would like to clarify that all our results are evaluated with DFT. In particular, energy above hull is reported based on DFT energies. We have added Appendix D.3 to make this point more clear. We only use a ML force field in some of our benchmarks to pre-relax the generated structures before evaluating them with DFT. This is useful to speed up DFT relaxation and allows us to pre-filter structures that we predict are not stable before running DFT. However, note that even without ML relaxation most generated structures are close to the DFT local energy minimum (Fig. 2(c)).

In addition, we have experimentally synthesized and characterized TaCr₂O₆, a material that was conditionally sampled by MatterGen, demonstrating that our model is able to generate novel, synthesizable materials (Section 2.6, Appendix D.10). We have also performed extensive computational experiments to evaluate the model’s ability to rediscover known compounds unconditionally (Appendix D.5.4), when conditioned on bulk modulus (Appendix D.10.2), and when conditioned on magnetic density and HHI score (Appendix D.9.3). These results provide further evidence that our model is able to generate synthesizable materials.

Q3.3 Novelty of materials: In a reply to the recent Nature paper by Szymanski et al. (DOI: 10.1038/s41586-023-06734-w), Leeman et al. (DOI: 10.26434/chemrxiv-2024-5p9j4) pointed out that novelty of materials is not well-defined, and a definition as used in this manuscript (“We consider a structure to be novel if it is not contained in Alex-MP-ICSD”) is certainly not enough.

We agree that defining novelty is a challenging problem on its own, especially for high-throughput materials predictions. As highlighted by Leeman et al. [6], it is important to check whether a potentially novel structure is in fact (a) an ordered approximation of an already known compositionally disordered structure, or (b) whether there already exists another ordered approximation that belongs to the same disordered structure as the predicted one. To address this issue, we have updated our definition of novelty (and uniqueness) as follows (Appendix D.3). First, we replace the previous structure matcher by a novel disordered structure matching algorithm inspired by the well-known Hume-Rothery rules [2] (Appendix D.4). Second, we have significantly expanded our reference dataset by including 117,700 disordered structures from ICSD. According to our benchmarks (Appendix D.4), the disordered structure matcher is highly effective at detecting compositional disorder. Consequently, we have updated all results in the manuscript to adopt our revised definition of novelty and uniqueness.

Q3.4 Novelty of methods: The authors mention several times that a novel diffusion process is introduced. Diffusion processes for the generation of crystal structures were introduced before, e.g. cited in [4]. Method-wise, the novelty here is limited to changes in the architecture of the diffusion model. The main difference is the size of the dataset, as the Alexandria dataset is added to the training. The Alexandria dataset is almost 10 times larger than the MP dataset and thus becomes the largest part of the combined dataset. No comparison of the previous model from [4] on the new dataset is presented, which makes it impossible to differentiate between improvements caused by increasing the size of the dataset and improving the model. I am aware that the new model introduced here has more capabilities than the model presented in [4], but common aspects of the results, mainly presented in Section 2.2, should be compared in more detail to allow reliable conclusions regarding the improvements.

Thank you for your feedback. We have updated the manuscript to clarify in which way the method is novel. In addition, we have added a new ablation study (Appendix A.10) showing that of the six model innovations we ablate, all of them have a significant impact on the model performance. Furthermore, we have added a discussion (Appendix A.11) that highlights the model innovations of MatterGen compared to CDVAE.

To disentangle the effect of performance improvements because of model innovations versus scaling up the dataset, in Fig. 2 we compare both MatterGen and MatterGen-MP with CDVAE. MatterGen-MP and CDVAE are trained on the same dataset (MP-20) and therefore any performance improvements must be due to model innovations. In contrast, any performance improvements from MatterGen-MP to MatterGen must be due to the data. As can be seen, both model innovation and data scaling contribute significantly to the performance.

Finally, we would like to stress that curating the Alex-MP-ICSD dataset is a significant contribution in itself. To generate this dataset, we have re-calculated all energies of structures in Alexandria and ICSD via DFT to be consistent with Materials Project settings, resulting in more than 700,000 additional DFT calculations. Further, the data has been carefully processed to exclude invalid or duplicate structures as well as radioactive elements (Appendix C.1). We will make the data publicly available as part of our data release upon publication of the manuscript.

Q3.5 *State-of-the-art: The authors compare their results to CDVAE [4], but not to the newer publication DiffCSP [55], which is mentioned in the manuscript but for the first time only on page 8 in Section 2.4 of the Results.*

Thank you for pointing this out. We have added a comparison with DiffCSP to the manuscript, showing that MatterGen-MP generates 60 % more S.U.N. structures while the average RMSD of the generated structures is 50 % lower (Fig. 2(e-f)). We have also trained DiffCSP on our large Alex-MP dataset and find that MatterGen outperforms DiffCSP by more than 5x on RMSD and 10 % on the percentage of S.U.N. structures (Table D4).

Q3.6 *Reference of unpublished work: "We equip all methods with the MatterSim [52] MLFF, which is used to pre-relax and filter the generated structures by their predicted stability before running more expensive DFT calculations.": [52] is a non-published paper, so it is impossible to judge on the quality and reliability of that work.*

A pre-print of MatterSim is now publicly available [9]. We have updated the reference accordingly.

Q3.7 *Code availability: The authors state that the MatterGen code will be made available upon the publication of the manuscript.*

We are sharing a preliminary code release for the purpose of this review. We are ready to release the code upon publication of the manuscript.

Q3.8 *Figure 2d: Why is the novelty rate dependent on the number of generated samples? Are samples generated repeatedly? How do the authors explain that, if the pool of potential structures is 10^{10} or more, as stated in the introduction? Do you assume that the model suffers from some form of mode collapse, or do you assume that the model is hitting the limit of all possible structures?*

The novelty rate does not depend on the number of generated samples since—as you pointed out—we are far away from exhausting all possible structures. We have adapted Fig. 2(d) to make this more clear. However, uniqueness is indeed dependent on the number of generated samples: the more samples are generated, the more likely it is that we generate the same structure more than once. We hypothesize that the main reason for this is the still limited dataset size, e.g., our training dataset of ca. 600,000 stable structures is still orders of magnitudes smaller than commonly used image datasets like ImageNet [4] (14M) or LAION-400M [8] (400M) for training image generative models. In addition, note that while two structures might look different from the model’s perspective, our strict uniqueness definition might still consider them to be the same. This is partially mitigated by the fact that we curate the training set to only contain unique structures according to our definition, but it is still non-trivial for a model to learn.

References

- [1] M. Aykol, S. S. Dwaraknath, W. Sun, and K. A. Persson. Thermodynamic limit for synthesis of metastable inorganic materials. *Science advances*, 4(4):eaq0148, 2018.
- [2] W. D. Callister, D. G. Rethwisch, A. Blicblau, K. Bruggeman, M. Cortie, J. Long, J. Hart, R. Marceau, and R. Mitchell. *Materials science and engineering: an introduction*, volume 7. John Wiley & sons New York, 2007.
- [3] C. Chen and S. P. Ong. A universal graph deep learning interatomic potential for the periodic table. *Nature Computational Science*, 2(11):718–728, 2022.
- [4] J. Deng, W. Dong, R. Socher, L.-J. Li, K. Li, and L. Fei-Fei. ImageNet: A large-scale hierarchical image database. In *2009 IEEE conference on computer vision and pattern recognition*, pages 248–255. IEEE, 2009.

- [5] R. Jiao, W. Huang, P. Lin, J. Han, P. Chen, Y. Lu, and Y. Liu. Crystal structure prediction by joint equivariant diffusion. *Advances in Neural Information Processing Systems*, 36, 2024.
- [6] J. Leeman, Y. Liu, J. Stiles, S. B. Lee, P. Bhatt, L. M. Schoop, and R. G. Palgrave. Challenges in high-throughput inorganic materials prediction and autonomous synthesis. *PRX Energy*, 3(1):011002, 2024.
- [7] N. Liu, S. Li, Y. Du, A. Torralba, and J. B. Tenenbaum. Compositional visual generation with composable diffusion models. In *European Conference on Computer Vision*, pages 423–439. Springer, 2022.
- [8] C. Schuhmann, R. Vencu, R. Beaumont, R. Kaczmarczyk, C. Mullis, A. Katta, T. Coombes, J. Jitsev, and A. Komatsuzaki. LAION-400M: Open dataset of clip-filtered 400 million image-text pairs. *arXiv preprint arXiv:2111.02114*, 2021.
- [9] H. Yang, C. Hu, Y. Zhou, X. Liu, Y. Shi, J. Li, G. Li, Z. Chen, S. Chen, C. Zeni, et al. MatterSim: A deep learning atomistic model across elements, temperatures and pressures. *arXiv preprint arXiv:2405.04967*, 2024.

Response to referees for manuscript “A generative model for inorganic materials design”

We thank all referees for their additional feedback and suggestions. We are glad that reviewers 1-3 found our response and revisions to have addressed all of their previous comments. We believe that the quality and clarity of the revised manuscript has improved significantly thanks to the referees’ feedback. Below, we provide point-to-point responses to all remaining comments as well as to the editorial suggestions. We highlight changes to the manuscript and supplementary information in blue.

Contents

1	Reviewer 1: rare earth free magnets discovery	1
2	Reviewer 2: diffusion models	2
3	Reviewer 3: AI materials discovery	2
4	Reviewer 4: synthesis and characterization	2
5	Editor	4

1 Reviewer 1: rare earth free magnets discovery

Overall I found that the revised manuscript to have addressed the major concerns that I had. There were substantial changes made to the manuscript and to the appendices. The author’s responses were complete.

Thank you for carefully reading the revised manuscript and supplementary information.

Q1.1 *In particular, there were substantial changes to appendix D on validating the approach with known compounds, experimental validation and improved clarity on the model and its limitations. The section on ferromagnetic compounds while validating the model also points out a limitation which I think should be noted but not a ‘fatal flaw’ due to the complexity of magnetic compounds. Specifically, their predication of the magnetic density of $\text{Ca}_1\text{Mn}_4\text{Fe}_3\text{O}_{12}$, which is experimentally observed to be antiferromagnetic thus ‘resulting in a magnetic density that is lower than if it was ferromagnetic’. When calculating the magnetic flux density, how the ferromagnetic ions couple is dependent on their location, ordering on sub-lattices, and presence of anions and/or interstitials which can affect their separations etc. So, when searching for functional materials like permanent magnets (PM), it is important to not only get the crystal structure but the magnetic structure as well. To be a commercially viable PM, the compound also needs to have high coercivity, which is related to its magneto-crystalline anisotropy (MCA). A large MCA however is a necessary but not a sufficient condition for high coercivity, there are extrinsic factors associated with the microstructure that control how much coercivity can be achieved. And of course, this all has to be in a temperature range needed by an application. So, while MatterGen I believe has made some nice inroads to this very complex challenge, there is still a lot that needs to be done. Pointing this out would help put this work in context of the larger challenge.*

Thank you for pointing this out. We agree that the problem of discovering functional permanent magnets is not solved, and that many other properties—both at the microscale and at the macroscale—would have to be computed and predicted to determine whether a candidate material is a viable permanent magnet.

To address this, we have revised Section 2.5 as follows: “We simplify this task to finding materials with a high magnetic density of 0.2 \AA^{-3} and a low Herfindahl–Hirschman index (HHI) score of 1250, where a material with an HHI score below 1500 is considered to have low supply chain risk [47] (experimental details in Supplementary D.9.1); **in practice, additional properties like high coercivity, suitable magnetic order and critical temperature need to be satisfied**”. Note that we already have a similar disclaimer in Section 2.4: “While these tasks were chosen to evaluate the model’s generality, further investigations would be required to assess the suitability of

these materials for specific applications, e.g., a superhard material needs to have a high shear modulus, and a permanent magnet needs a suitable magnetic order and critical temperature”.

In addition, we have expanded Supplementary D.9.3: “In general, it may be that other magnetic orderings, such as the antiferromagnetic ordering, are lower energy and are hence more likely to be synthesizable than the ferromagnetic one. This suggests that a model for generating functional permanent magnets must be able to successfully predict the most stable magnetic ordering. Likewise, generated magnets would have to satisfy additional properties of interest such as a suitable critical temperature and high coercivity, which is related to its magneto-crystalline anisotropy. We acknowledge that these are complex properties that depend on many factors, which makes these properties difficult to simulate and predict. Solving these challenges would require substantial effort that is beyond the scope of this work.”

2 Reviewer 2: diffusion models

I had provided a detailed and extensive review of the initial version of the manuscript. I have read the authors’ reply to this initial review and looked at the revised manuscript. My raised questions and concerns have been addressed in a satisfactory manner, and sufficient modifications have been implemented in the manuscript. I also appreciate the additional evaluations and the experimental synthesis of some materials, which provide further value. The use of statistics seems appropriate and error bars, when shown, are explained in the figure caption. Overall, I support publication of the manuscript.

Thank you for reviewing the revised manuscript again after your detailed initial review. We appreciate your support for publication.

3 Reviewer 3: AI materials discovery

Thank you very much for substantially extending the manuscript and carefully replying to all comments of the three reviewers. My main concerns are addressed very well: 1) Stability concerns are discussed and explained well 2) Additional novelty metrics are introduced 3) Additional ablation studies are introduced to disentangle data from model improvements 4) Additional baselines (DiffCSP) are introduced 5) Minor comments are addressed in a satisfying way 6) Code availability: The authors sent the code to the reviewers and will release it after the publication of the manuscript. This needs to be checked by the editorial team. Overall, these improvements significantly increase the quality of the article. Thus, I support publication in Nature.

Thank you for your feedback. We agree that your comments have helped to increase the quality of the work, and we appreciate your support for publication.

4 Reviewer 4: synthesis and characterization

Thank you for taking the time to read the revised manuscript and supplementary information as well as for the helpful comments. We give a point-by-point response below.

Q4.1 *In $TaCr_2O_6$, if the valence of Cr is 3+, that of Ta should be 6+, which is not a stable state.*

Thank you for your comment. We have performed an x-ray spectroscopy (XPS) analysis on the synthesized $TaCr_2O_6$ sample to assess the valence states of its elements. We analyze the observed XPS spectra in the newly added Supplementary D.10.7 and provide the raw XPS measurements as part of our data release.

In summary, we observe the presence of Ta^{5+} , Cr^{3+} , and Cr^{4+} which can result in a balanced $TaCr_2O_6$ structure where half the Cr atoms have a 3+ valence state and the other half a 4+ valence state (Fig. 1). This result is in line with our XRD spectra measurements, providing additional experimental evidence that the synthesized compound is indeed a compositionally disordered phase of $TaCr_2O_6$ in a rutile phase, as stated in the manuscript.

Q4.2 *Cr_2MoO_6 : $MoCr_2O_6$, with trirutile type crystal structure, was synthesized at high pressure and high temperature (Journal of Solid State Chemistry, 1978, 23 [3-4], 315-319). Therefore, it is not surprising that the authors could not synthesize Cr_2MoO_6 at ambient pressure in air.*

Thank you for pointing this out. We have added a citation in Supplementary D.10.5 regarding the existence of an experimentally synthesized high-pressure phase of Cr_2MoO_6 . We also note that our structure matcher identified the MatterGen-generated structure as novel because the previously synthesized crystal structure is not included in ICSD, and is therefore not present in our reference dataset.

Figure 1: **XPS measurements for the TaCr₂O₆ powder sample.** (a) Full XPS measurement for the sample, with annotation of major peaks. (b-d) XPS analysis for the Cr2p, O1s, and Ta4f peaks, respectively. The black circles indicate the measured electron count, the black lines indicate the fitted electron count, and the colored Gaussians highlight the single peaks. Source plots are obtained from the Thermo Fisher Avantage software. Vertical gray lines act as a visual aid to highlight the binding energy of each fitted peak.

We have added the following sentence to Supplementary D.10.5: “We note that Cr₂MoO₆ has been previously experimentally synthesized at high temperature and pressure in [1]. The MatterGen-generated structure was identified as novel because the experimental entry is not listed in ICSD, and is therefore not included in our reference dataset.”

Q4.3 In LaMoO₄, if the valence of Mo is 6+, that of La should be 2+, which is not a stable state.

We agree that it is possible that the structure was not successfully synthesized because of the uncommon valence state the La atoms should assume. We did not further analyze the sample given that the synthesis was clearly unsuccessful.

Q4.4 In Mn₃NiO₆, the valences of Mn and Ni should be 3+, while Ni with valence of 2+ is stable at high temperature. Therefore, it is reasonable that the authors obtained NiMn₂O₄ with spinel structure, while manganese oxide should be also observed in the XRD pattern.

Thank you for the observation. While your explanation is plausible, we did not analyze the XRD pattern of the sample in detail because it clearly indicated an unsuccessful synthesis of the desired compound.

Q4.5 When using high energy ball milling, it is important to state conditions/parameters, such as milling media, materials of vial/balls, diameter of balls, ball-to-powder weight ratio, quantity of water/alcohol used, rotation speed,

Thank you for highlighting this point. We have added conditions and parameters for the high energy ball milling process in Supplementary D.10.4: “A high-energy planetary ball mill (Model: QM-3SP2) was employed to grind the mixture of Ta₂O₅ and Cr powders. The milling process utilized a 100 mL zirconia jar paired with zirconium oxide grinding beads, employing a mix of 3 mm and 5 mm beads in a ratio of 2:1. This mixed approach leverages the individual benefits of both larger and smaller beads, thereby optimizing grinding efficiency, particle size uniformity, and distribution. The bead-to-material ratio was maintained at 1:1.5, ensuring a sufficient amount of grinding media relative to the powder, while keeping approximately one-third of the jar’s volume empty to facilitate proper milling dynamics. The mixture underwent high-energy milling at 500 rpm in alternating

directions, with 1-hour intervals in each direction. This forward-reverse sequence was repeated for a total of 12 cycles to facilitate uniform particle size reduction and thorough mixing of the material components.”

We then refer to the above details for other synthesis attempts in Supplementary D.10.5: “The mixture was processed using a high-energy planetary ball mill (model: QM-3SP2), following the procedure detailed in Supplementary D.10.4.”

5 Editor

Thank you for your editorial suggestions. Please find point-to-point responses to the suggestions below.

TITLES *Titles cannot exceed 75 characters (including spaces); they must not contain punctuation.*

We have removed punctuation from the title and made sure the title does not exceed 75 characters.

SUMMARY PARAGRAPH *All Nature papers begin with a fully referenced paragraph, typically no longer than 200 words, aimed at readers in other disciplines. This paragraph starts with a 2- to 3-sentence, basic introduction to the field; continues with a 1-sentence statement of the main findings starting ‘Here we show’ or an equivalent phrase; and finally, concludes with 2 to 3 sentences putting the main findings into general context so it is clear how the results described in the paper have moved the field forward. A downloadable, annotated example is available at <https://www.nature.com/nature/for-authors/formatting-guide>. In some cases it may be necessary to exceed this limit in order to explain complex material for readers in other fields – in such cases, summary paragraphs can be up to 230 words in length. The extra length, however, is for introduction and context, and not for additional technical information.*

We have shortened the summary paragraph to less than 200 words, following the suggested structure.

MAIN TEXT *If further introductory material is necessary, the main text can begin with up to 500 words of introduction expanding on the background to the work (some overlap with the summary paragraph is acceptable), before proceeding to a concise, focused account of the new research and findings, and ending with 1 or 2 short paragraphs of discussion. Sections are separated with subheadings to aid navigation. Subheadings may be up to 40 characters (including spaces).*

We have shortened the introduction, discussion, as well as subheadings.

STATISTICS *Authors should ensure that any statistical analysis used is sound and that it conforms to the journal’s guidelines (see <https://www.nature.com/nature/for-authors/formatting-guide> for guidance).*

We have double-checked that our statistical analyses are sound and conform to the journal’s guidelines.

METHODS *At the end of the main text document (after the main figure legends), there should be a section entitled “Methods”, which provides a more detailed discussion of the additional methodological information that would allow other researchers to replicate the results (we define “Methods” quite broadly, so this is not limited to details of experimental protocols – supplementary discussion and analysis can also be included). The Methods section will not appear in the print version but will be fully copy-edited and appear online in the full-text HTML and PDF versions. The Methods section should be written as concisely as possible but should contain all elements necessary to allow interpretation and reproduction of the results. If there are additional references in the Methods section, their numbering should continue from the last reference in the main paper, and the list should follow the Methods section. If the methods require chemical structures, figures or tables, these should be supplied as Extended Data (see below). For mathematically complex methods, or methods that require an unusually large number of figures or tables (beyond what can be accommodated as Extended Data), the entire Methods section should instead be supplied as a separate Supplementary Information.*

Given the complexity of our methods and the length of our supplementary material, we provide the entire Methods section as a separate Supplementary Information.

REFERENCES *As a guideline, Articles allow up to 50 references in the main text; additional references can be cited in (and listed after) the Methods section, as detailed above.*

We have reduced the references in the main text to 50.

MAIN TEXT STATEMENTS We require authors to provide a detailed Author Contribution statement immediately after the acknowledgements; the specific contributions of each author must be listed. It is also a condition of publication that authors include an Author Information statement indicating how to access information regarding reprints and permissions, stating whether or not there is a financial or non-financial competing interest, and naming the author to whom correspondence and requests for materials should be addressed. Please ensure that this section is included in the manuscript file after the Methods (but before the Extended Data legends) - it will not appear in the print version but will appear online in the full-text HTML and PDF versions. For details of "end note" style and an example see <https://www.nature.com/nature/for-authors/formatting-guide>.

We have added/modified the main text statements accordingly.

DATA AVAILABILITY STATEMENT All published manuscripts reporting original research in Nature Portfolio journals must include a data availability statement. The data availability statement must make the conditions of access to the "minimum dataset" that are necessary to interpret, verify and extend the research in the article, transparent to readers. This minimum dataset may be provided through deposition in public community/discipline-specific repositories, custom proprietary repositories for certain types of datasets, or general repositories like Figshare, Zenodo and Dryad. Providing large datasets in supplementary information is strongly discouraged and the preferred approach is to make data available in repositories. More information on Nature Portfolio's reporting standards and preparing your Data Availability Statement can be found here: <https://www.nature.com/nature-portfolio/editorial-policies/reporting-standards#reporting-requirements>. For all studies using custom code or mathematical algorithm that is deemed central to the conclusions, a statement must be included under the heading "Code availability", indicating whether and how the code or algorithm can be accessed, including any restrictions to access. Code availability statements should be provided as a separate section after the data availability statement but before the References. Code should be deposited in a DOI-minting repository such as Zenodo, Gigantum or Code Ocean and cited in the reference list. Authors are encouraged to manage subsequent code versions and to use a license approved by the open source initiative. Additional details can be found here: <https://www.nature.com/nature-research/editorial-policies/reporting-standards#availability-of-computer-code>.

We have added data and code availability statements.

FIGURE LEGENDS These should be listed sequentially after the references in the main text and not in the figures files. Each legend should begin with a brief title for the whole figure and continue with a short description of each panel and the symbols used. Any error bars in the figures must be defined (for example, s.d., s.e.m.) and the value of n indicated; see <https://www.nature.com/nature/for-authors/formatting-guide> for further explanation.

We have double-checked that the figure legends follow the guidelines and that error bars are clearly explained.

DISPLAY ITEMS We ask that you take stock of all the data that have been generated throughout the review process and ensure that only the data most central to the conclusions are presented in the main figures. Figures should be comprehensible to readers in other or related disciplines, and assist their understanding of the paper. We encourage authors who are describing complex processes to include a schematic of the main finding as part of the Extended Data to aid readers unfamiliar with the immediate discipline. Figures should be as small and simple as is compatible with clarity. All panels of a figure should be logically connected; each panel of a multipart figure should be sized so that the whole figure can be reduced by the same amount and reproduced on the printed page at the smallest size at which essential details are visible. For guidance, Nature's standard figure sizes are 89 mm (one column), 120 mm (one and a half columns), or exceptionally 183 mm (two columns) wide; the full depth of a Nature page is 247 mm. All panels of figures should be presented on a single page and assembled into a rectangular shape for publication; please indicate any essential alignments (parts horizontal, vertical, spacings of stereo pairs, etc.). Tables should be prepared using the Table menu in Microsoft Word.

We have double-checked that the display items are comprehensible, compact and simple.

CHEMICAL STRUCTURE PRESENTATION Any chemical structures in the main text or Extended Data figures must conform to our chemical structure style guide (<https://www.nature.com/documents/nr-chemical-structures-guide.pdf>). This guide lists the ChemDraw preferences and stylesheet (<https://www.nature.com/documents/nr-chemdraw-stylesheet.cds>) that must be used to draw all structures. The style and size of chemical structures should not be modified from the default settings in the template, unless absolutely necessary (see the guide for examples), in which case 80% size and 5pt font is the smallest size possible. Please

export any ChemDraw (.cdx) files as a PDF, retaining editing capabilities — we find that ‘print to pdf’ works well for this — and upload this alongside your manuscript.

Not applicable. However, in Supplementary D.2.1 we explicitly specify how crystal structures are visualized.

FIGURE FORMATTING *Lettering in all figures (labelling of axes and so on) should be in uniform, sans-serif font, in lower-case type, and large enough to permit substantial reduction for publication (minimum font size 5 pt). Separate parts of a figure are labelled a, b, etc. Units have a single space between the number and the unit, and follow SI nomenclature or the nomenclature common to a particular field. Thousands are separated by commas (1,000). Unusual units or abbreviations are defined in the legend. Scale bars rather than magnification factors should be used.*

We have double-checked that the figures are formatted accordingly.

IMAGE PRESENTATION *Authors should be aware that any image provided for publication, either in print or online (as Extended Data or Supplemental Information), may be subject to a quality control process to check for image integrity and manipulation. For a full discussion of our standards regarding how images should be prepared and presented, see www.nature.com/authors/editorial_policies/image.html.*

We acknowledge that any image provided may be subject to a quality control process.

EXTENDED DATA *Nature is now integrating the supplemental figures and tables into the final version of most papers. Extended Data do not appear in the printed version of the paper but are included online within the full-text HTML and at the end of the online PDF. Extended Data are an integral part of the paper and only data that directly contribute to the main message should be presented. All Extended Data must be referred to in the main text, figure legends and/or Methods section, and their figure legends should be listed sequentially at the end of the main text, not in the Extended Data files. Authors should assemble the Extended Data into a maximum of ten, A4 size, multi-panelled display items, submitted as individual JPEG, TIFF or EPS files. They must be provided at the same quality as figures for print, but there are important differences in their formatting. More specific instructions are provided in the Extended Data Formatting Guide (http://s3-service-broker-live-19ea8b98-4d41-4cb4-be4c-d68f4963b7dd.s3.amazonaws.com/uploads/ckeditor/attachments/7823/3h_Extended_data.pdf).*

Not applicable.

SUPPLEMENTARY INFORMATION *Supplementary Information is online-only material published with the manuscript (<https://www.nature.com/nature/for-authors/supp-info>). For most papers, there should be no need for Supplementary Information beyond that already provided as Methods and Extended Data, the aim being to avoid unnecessary fragmentation of the paper online. Exceptions to this rule include large datasets that cannot be accommodated within Extended Data; video material; and more complex “Supplemental Methods” (and any associated references) that do not readily fit within the constraints of the Methods/Extended Data formats. Please note that after the paper has been formally accepted you can only provide amended Supplementary Information files for critical changes to the scientific content, not for style. You should clearly explain what changes have been made if you do resupply any such files.*

As mentioned above, we include Supplementary Information given the complexity of the method and the length of the supplementary material.

SOURCE DATA *To further increase transparency, we encourage authors to provide, in spreadsheet form, the data underlying the graphical representations used in the figures. This is in addition to our well-established data-deposition policy for specific types of experiments and large datasets. Readers of the online manuscript will be able to access the Source Data directly from the figure legend. Spreadsheets can only be submitted in .xls, .xlsx or .csv formats. One file per figure is permitted; thus, if there is a multi-panelled figure the Source Data for each panel should be clearly labeled in the csv/Excel file; alternatively the data for a figure can be included in multiple, clearly labeled sheets within an Excel file. File sizes of up to 30 MB are permitted; however, it is expected that the vast majority of Source Data files will be considerably smaller than this. When submitting these files with your manuscript, please select the file type “Source Data” and use the title field in the file description tab to indicate the figure to which the Source Data pertain.*

Not applicable. However, we do include measurement data from XRD and XPS in our data release.

THIRD PARTY RIGHTS While you finalise your manuscript, please identify any content used in your article — whether in the main text, extended data or supplementary information — that comes from a third party. This could include figures, tables, images, videos or text boxes that are reproductions or adaptations of items that have previously been published elsewhere and/or are owned by a third party. It also encompasses pictures taken by professional photographers, maps and images downloaded from the internet. You must obtain the right to use each of these items and you will need to provide evidence that you have these rights before your paper can be accepted for publication. You will also need to give proper attribution to the copyright holders in your paper. Failure to obtain the appropriate rights will delay the publication of your article. Please fill out a Third Party Rights Table, and upload this to our manuscript tracking system with your manuscript. You can find more information about third party rights on our *Rights and permissions* page.

Not applicable.

References

- [1] A. Collomb, J. Capponi, M. Gondrand, and J. Joubert. Synthese de quelques oxydes mixtes de type A6+ B23+ O6 en milieu hydrothermal sous tres haute pression. *Journal of Solid State Chemistry*, 23(3-4):315–319, 1978.

We thank the reviewers for their constructive feedback. We provide point-by-point responses to the remaining comments below.

Referee #4 (Remarks to the Author):

1) Regarding TaCr₂O₆, the authors have indicated that Cr is present with both 3+ and 4+ valences simultaneously, which explains the given composition. But, Cr⁴⁺ is not very commonly observed.

Thank you for your comment. Our XPS analysis have indicated the presence of Cr in both 4+ and 3+ valence states. CrO₂ is a well-known compound in Rutile structure. Therefore, it is not surprising that Cr⁴⁺ exists in our synthesized TaCr₂O₆.

(2) Noting that the rest three samples are not successfully synthesized, why did not the authors try other compositions?

While we have started with four different synthesis targets, our goal for this study was to synthesize a single compound as a proof of concept. We are aware that synthesizing and characterizing materials requires significant efforts, so we have made a conscious decision to focus our efforts on the synthesis target that looks most promising.

Our synthesis is only a proof of concept, showcasing that MatterGen generated materials can be experimentally synthesized and has property close to our target. We are not claiming that our synthesized material is a useful material with practical applications. We've added notes that the synthesis is "a proof of concept" in our abstract, introduction section, and experimental section in our main text.